# A high-resolution two-step evolution experiment in yeast reveals a shift from pleiotropic to modular adaptation

Grant Kinsler[1,2]*, Yuping Li[3,4], Gavin Sherlock[3], Dmitri A. Petrov[1]

1 Department of Biology, Stanford University, Stanford, California, United States of America, 2 Department of Bioengineering, University of Pennsylvania, Philadelphia, Pennsylvania, United States of America, 3 Department of Genetics, Stanford University, Stanford, California, United States of America, 4 Department of Microbiology & Immunology, University of California, San Francisco, San Francisco, United States of America

☯ These authors contributed equally to this work.
* grantkinsler@gmail.com

**Data Availability Statement:** All relevant data are within the paper and its Supporting information files. Raw sequencing data are available at the

## Abstract

Evolution by natural selection is expected to be a slow and gradual process. In particular, the mutations that drive evolution are predicted to be small and modular, incrementally improving a small number of traits. However, adaptive mutations identified early in microbial evolution experiments, cancer, and other systems often provide substantial fitness gains and pleiotropically improve multiple traits at once. We asked whether such pleiotropically adaptive mutations are common throughout adaptation or are instead a rare feature of early steps in evolution that tend to target key signaling pathways. To do so, we conducted barcoded second-step evolution experiments initiated from 5 first-step mutations identified from a prior yeast evolution experiment. We then isolated hundreds of second-step mutations from these evolution experiments, measured their fitness and performance in several growth phases, and conducted whole genome sequencing of the second-step clones. Here, we found that while the vast majority of mutants isolated from the first-step of evolution in this condition show patterns of pleiotropic adaptation—improving both performance in fermentation and respiration growth phases—second-step mutations show a shift towards modular adaptation, mostly improving respiration performance and only rarely improving fermentation performance. We also identified a shift in the molecular basis of adaptation from genes in cellular signaling pathways towards genes involved in respiration and mitochondrial function. Our results suggest that the genes in cellular signaling pathways may be more likely to provide large, adaptively pleiotropic benefits to the organism due to their ability to coherently affect many phenotypes at once. As such, these genes may serve as the source of pleiotropic adaptation in the early stages of evolution, and once these become exhausted, organisms then adapt more gradually, acquiring smaller, more modular mutations.

Short Read Archive under BioProject Number: PRJNA1098711. Additional data including processed are available on Zenodo: https://zenodo.org/records/13336585. All yeast strains are available upon request.

**Funding:** Funding was provided by the National Institutes of Health (NIH R35GM11816506 awarded to DP, NIH R35GM131824 awarded to GS), funding from the Chan Zuckerberg Initiative awarded to DP, and National Science Foundation (NSF) grant no. 2225088 awarded to DP. GK received support of a CEHG predoctoral fellowship. The funders had no role in study design, data collection and analysis, decision to publish, or preparation of the manuscript.

**Competing interests:** The authors have declared that no competing interests exist.

## Introduction

As organisms adapt to their environment, they face a multidimensional optimization problem. To be advantageous, new mutations must improve one or more traits under selection without imposing strong costs on other traits. Theoretical analyses of adaptive walks in multidimensional trait spaces suggest that mutations that generate small phenotypic shifts in few traits are more likely to be beneficial overall than mutations of large phenotypic effect on many traits [1]. Consequently, adaptive mutations are expected to both provide small fitness benefits and to be **modular**—that is, affect only a few traits without affecting others.

Despite these theoretical expectations, empirical evidence suggests that mutations are not necessarily modular nor small in their effect on fitness. Studies of segregating variants in populations and those that arise in evolution experiments have found that pleiotropic variants can have large effects on traits and fitness [2–4], though the extent to which these variants improve, rather than harm, multiple traits is often not characterized. Microbial evolution experiments have revealed that early adaptation often proceeds by single mutations that provide large fitness benefits [5–9]. In the cases where the improvement of these mutations has been decomposed into distinct trait performances, it has been observed that these mutations can improve multiple traits simultaneously (as illustrated in Fig 1A) [5,10–12].

The observation of adaptive mutations improving multiple performances at once, which we here term "**pleiotropic adaptation**," can be easily seen in a series of evolution experiments conducted with barcoded yeast in which a comprehensive set of adaptive mutations was profiled for their effects on likely orthogonal trait performances [5,7,9,10]. Li and colleagues [10] in particular showed that approximately 85% of first-step adaptive mutations isolated from their evolution experiment improve performance in both fermentation and respiration growth phases, both of which are under selection during the yeast growth cycle. These pleiotropic mutants from this initial step of adaptation, many of which harbor only a single mutation in the Ras/PKA pathway, are also strongly adaptive, providing fitness benefits of up to 120% per

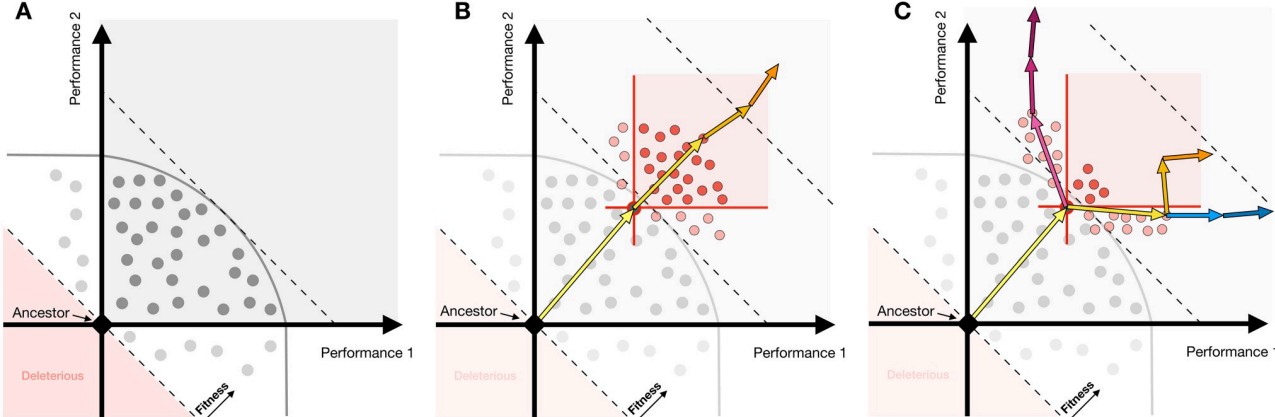

**Fig 1. Theoretical illustration: Pleiotropy may be a generic feature of adaptation or specific to the first-step of evolution. (A)** First-step adaptive mutations (each mutation depicted as a dot) in evolution often exhibit patterns of pleiotropic adaptation—improving performance in multiple traits simultaneously (falling into the gray square). Gray curved line represents the limits of combinations of performances reached by the first-step of evolution. **(B)** If pleiotropic adaptation is common, then second-step adaptive mutations (depicted in red) would continue to improve multiple performances at once. Longer adaptive walks would also continue to show these patterns (orange arrows). **(C)** If pleiotropic adaptation is rare and largely constrained to the first adaptive step, then second-step adaptive mutations might show a shift in their improvement, instead primarily improving one performance or the other (light red circles). In this scenario, longer term adaptive walks may continue to specialize in one performance or the other (depicted by blue and magenta arrows), or instead continue to collectively improve both performances, albeit in a stepwise manner (orange arrows).

growth cycle (roughly 15% per generation) [9]. Such large-effect Ras/PKA pathway mutations are commonly found in early evolution in other systems, such as cancer progression [13]. As the study we present here follows on from our previous series of findings, we use these yeast mutations as a motivating example throughout the rest of the introduction.

How do we reconcile our observations of pleiotropic adaptation [10] with theoretical expectations that these large-benefit mutations should affect only a small number of traits? One possibility is that fermentation and respiration performances are not as distinct as we believe and that these 2 modes of metabolism belong to the same physiological module [2,3,14]. However, these growth phases have distinct transcriptional, proteomic, and metabolomic profiles [15–18], and, moreover, a number of adaptive mutations in our series of yeast experiments do improve only one of these performances, demonstrating that it is in fact possible to shift one performance without affecting the other.

The other possibility is that, while these growth performances are not physiologically or genetically linked in general, the Ras/PKA pathway is wired in such a way that mutations that target this pathway are capable of being both pleiotropic and adaptive, affecting many phenotypes of the organism but in a coherent and coordinated fashion. Indeed, if mutations in this pathway are special, it might be that mutations in general might not have these patterns of pleiotropic adaptation and instead exhibit "**modular adaptation**," improving only a subset of the traits under selection. Thus, isolating and characterizing the effects of subsequent mutations, which may be less likely to target this already-mutated pathway, might better reflect the pleiotropic properties of adaptive mutations beyond these extremely beneficial first-step mutations in the Ras/PKA pathway. One way in which we can assess whether the observed adaptive pleiotropy is a common feature of adaptive mutations is to conduct adaptive walks, evolving populations further in the same environment. We can then ask whether later adaptive mutations continue to show pleiotropic adaptation or not.

The first possibility is that pleiotropic adaptation is indeed common. This may be true if there are many pathways in the cell that can be mutated to yield simultaneous improvement of the traits under selection or, instead, if the signaling pathways mutated early can continue to be optimized beyond the first adaptive step. In this scenario, second-step adaptive mutations would continue to improve both traits under selection (Fig 1B, red points) and longer adaptive walks would also continue to show this pattern of pleiotropic adaptation (Fig 1B, arrows).

Alternatively, pleiotropic adaptation may be rare, and first-step mutations in the Ras/PKA pathway target the only (or one of few) signaling pathway(s) which can result in simultaneous improvement of multiple traits (performance in both fermentation and respiration growth phases in the case of the yeast evolution experiments). For adaptation to continue, it would need to engage the modules that can independently control the performance in each growth phase. Individual second-step mutations under this scenario would then be expected to exhibit a pattern of "modular adaptation," improving only 1 performance under selection or the other (Fig 1C). The longer adaptive walks could continue down this route of specialization in either a single performance (blue or magenta arrows) or instead improve both performances under selection, but via sequential improvement of one performance and then the other (orange arrows).

Thus, to characterize the nature of single-step adaptive mutations and whether the observation of pleiotropic adaptation of first-step mutations is a general feature of individual adaptation-driving mutations or instead a rare feature of early adaptive mutations, we need to experimentally conduct high-resolution adaptive walks, wherein we can isolate adaptive mutations, quantify their effects on traits relevant to fitness, and identify the molecular basis of adaptation. The yeast barcoding system developed by Levy and colleagues [7] is particularly well-suited for this set of experiments, as we can isolate hundreds

of mutations per evolution experiment and study their properties via pooled fitness measurement experiments.

In this study, we perform second-step evolution experiments using a set of 5 first-step adaptive mutations isolated from a glucose-limited evolution experiment [7] as new ancestors. We then isolate hundreds of mutants from these evolution experiments and measure their performance in the growth phases that make up the evolution condition. We find a shift in the nature of adaptation over this two-step adaptive walk. While first-step mutations primarily demonstrate pleiotropic adaptation, improving performance in both growth phases under selection, second-step mutations instead primarily exhibit modular adaptation, improving performance in only a single growth phase under selection (Fig 1C). Whole genome sequencing reveals an associated shift in the molecular basis of adaptation: from first-step mutations in general signaling pathways to second-step mutations in genes related to mitochondrial function and respiration. Finally, we sample rare adaptive clones that showed patterns of adaptive pleiotropy and discovered that they harbor multiple additional mutations. This suggests that these populations have not yet reached physiological constraints, which prevent further improvement of the growth phases measured here. Instead, these data suggest that adaptation may be constrained by the structure of genetic modules which prevent adaptive mutations from improving multiple performances in a single step.

This shows that early adaptation, here represented by the first-step in our evolution experiment, can engage signaling pathways that allow for rapid, large step pleiotropic adaptation but later adaptation may be more likely to be modular, as predicted by theory. We thus expect that longer term evolution will indeed progress through smaller, and ultimately more modular, adaptive mutations.

## Results

### Isolating second-step adaptive clones and measuring performance in growth phases

When yeast are grown in an environment under glucose-limitation in batch culture, they experience several growth phases (Fig 2A). First, the yeast experience lag phase, where they acclimate to the environment and allocate cellular resources to consuming glucose. Then, the yeast ferment the glucose, converting it to ethanol. Once the glucose is consumed, the yeast then undergo the diauxic shift and respire on the ethanol they produced during fermentation. Finally, once the supply of ethanol has been depleted, the yeast experience stationary phase, where they allocate resources to surviving without a carbon source. These growth phases are typically thought of as independent processes, with distinct transcriptional, proteomic, and metabolomic profiles that characterize and drive yeast physiology [15–18].

Previously, a population of barcoded yeast was evolved in a 2-Day transfer environment under glucose limitation, where they experienced lag, fermentation, and respiration but not stationary phases before being transferred to fresh medium [7,9]. Adaptive mutations isolated from this experiment gained substantial fitness benefits, primarily by constitutively activating one of 2 glucose-sensing pathways: Ras/PKA and TOR/Sch9 [9]. Additionally, 85% of these mutants improved performance in both fermentation and respiration phases, despite the supposed independence of these growth phases. Interestingly, with additional evolution experiments designed to maximize individual performances, Li and colleagues [5] were able to find evidence of constraints on the first step of evolution such that no single mutation is able to simultaneously maximize both fermentation and respiration performances to the largest extreme of each performance observed individually.

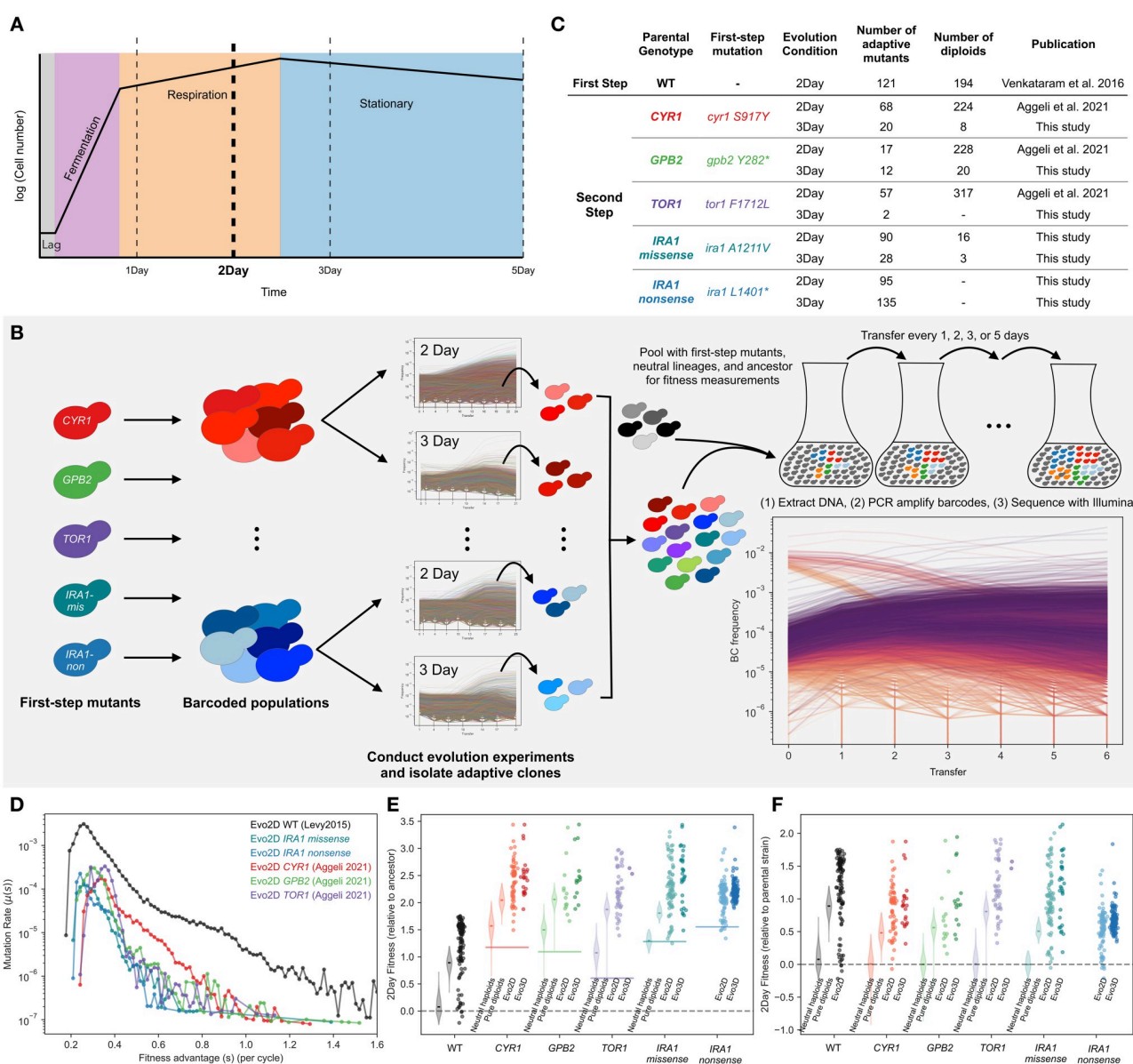

**Fig 2. Summary of experiments and fitness effects of isolated adaptive mutants. (A)** A schematic of yeast growth phases under the nutrient conditions used in this study. The yeast experience 4 h of lag phase, 16 h of fermentation, and 4 h of respiration phase in the first 24 h of growth. **(B)** Schematic of barcoded evolution experiments and fitness measurement experiments. **(C)** Table of mutants used in this study, including ploidy, and publication source. **(D)** Probability density of mutational fitness coefficients relative to each parental strain. The black line refers to first-step mutants from Levy and colleagues [7]. Colored lines depict the inferred density of fitness effects of mutations from second-step evolution experiments in the 2-Day transfer environment (Evo2D). **(E)** Fitness effects per cycle in 2-Day transfer of all mutants, relative to WT ancestor. First violin plot for each parental strain shows neutral haploids. Second shows pure diploids. Third column is all other 2-Day adaptive mutants, including adaptive haploids and high-fitness diploids. Fourth column is all other 3-Day adaptive mutants. **(F)** As in (E), but relative to parental strain. The data and code underlying this figure can be found in https://zenodo.org/records/13336585.

To understand whether pleiotropic adaptation is common or if instead first-step mutations represent rare solutions that improve both traits under selection, we carried out second-step evolution experiments in the same 2-Day transfer environment, isolated adaptive mutants, identified causative mutations underlying adaptation, and characterized the mutations' effects on performance in the environment's growth phases. Aggeli and colleagues [19] previously

performed second-step evolution experiments using barcoded populations that carried one of 3 mutations identified in the first step of evolution: a gain-of-function mutation in *CYR1*, a loss-of-function mutation in *GPB2*, and a gain-of-function mutation in *TOR1*. Here, we used additional barcoded populations derived from 2 distinct mutations in *IRA1*: one missense mutation and one nonsense mutation (see Methods). We then evolved 2 replicates of each barcoded population in the 2-Day transfer condition, labeled here "Evo2D," for 22 transfers (approximately 176 generations) and isolated adaptive clones (Fig 2B and 2C). As we were also interested in how the number of traits under selection alters the extent of pleiotropic adaptation, we also evolved the same barcoded populations in a 3-Day transfer condition, herein termed "Evo3D," where populations experienced an additional 12 h of respiration and 12 h of stationary phase, and isolated adaptive clones from this additional set of evolution experiments (Fig 2A–2C, see Methods).

To assess the extent to which physiological and genetic constraints affect the second-step of adaptation, we quantified each mutant's performance in fermentation, respiration, and stationary growth phases using pooled barcoded fitness assays, as developed previously (Fig 2B; [5,9,10,20,21]). Briefly, we pooled all isolated second-step mutants together with the barcoded mutants from the first step of evolution. We then mixed this pool of barcoded yeast with a set of barcoded neutral lineages and the ancestral strain, such that the barcoded pool started at either 2% or 5% frequency in the population, the neutral barcoded lineages collectively represented 2% of the population, and the ancestral strain made up the remaining >90% of the population (see Methods). We then measured the fitness of each mutant by serially transferring approximately $5 \times 10^7$ cells for 5 cycles in 1-, 2-, 3-, and 5-Day transfer intervals. At each transfer, we froze down the remaining cells, extracted their DNA, amplified the barcode region with PCR, and then sequenced the barcode region. We then calculated each mutant's fitness relative to the ancestor by comparing each mutant's frequency change with the pool of neutral lineages (Fig 2B, see Methods).

During the analysis of fitness measurement data, we observed that the detected fitness effects of each mutant varied systematically over the course of serial transfers during the fitness measurement of the isolated adaptive clones. Specifically, in the 2-Day transfer condition, many adaptive mutants showed very high fitness when the ancestral strain was at or above 80% of the population but showed much lower fitness at later time intervals when the pool of adaptive lineages dominated the population. We note that this effect is not due to change in population mean fitness, as this is already accounted for in these fitness values. While intriguing, we avoided these frequency-dependent fitness effects in our data by using only early time points, where the ancestor dominated the population, as these reflect the fitness in the environment set by the ancestor and where the fitness of mutants isolated from the original evolution experiment matched their fitness measurements in previous experiments (S1 Fig).

## Second-step adaptive mutations provide substantial yet smaller fitness benefits than first-step mutations

We sequenced the barcodes in these populations to monitor the dynamics of evolution and to quantify the distribution of fitness effects. Using the approach implemented in software Fit-Mut1 [7,22], we quantified the distribution of fitness effects for these populations and the original evolution experiment. Because auto-diploidization is a common mode of adaptation in evolution experiments with haploid yeast, we also used a benomyl assay to determine the ploidy of the isolated adaptive clones (Fig 2C). We then categorized mutants according to their ploidy status and fitness across pooled fitness measurement experiments as neutral haploids,

pure diploids, adaptive haploids, or high-fitness diploids (diploids that have additional beneficial mutations, see Methods).

We found that the rate of beneficial mutations is reduced in the second step of evolution in the 2-Day environment, with adaptive mutations that provide fitness benefits of 1.0 or greater (per cycle; approximately 8 generations per cycle) becoming much rarer (Fig 2D). This is consistent with the patterns of diminishing returns epistasis commonly observed in microbial evolution experiments [6,8,19,23–26]. While we see this general decrease in the magnitude of the fitness benefit of adaptive mutations, we nonetheless find that many second-step adaptive mutations still have substantial fitness gains in the 2-Day transfer evolution condition. Across all isolated second-step adaptive mutants (excluding auto-diploids and neutral haploids), the average fitness benefit provided is 82% per cycle relative to the parental strain. This is similar across mutants isolated from both 2- and 3-Day evolution experiments (Fig 2E and 2F). We also sampled rare mutants with fitness advantages as high as or even higher than the most extreme fitnesses observed in the first step of evolution. For example, 2 mutants isolated from the Evo3D *IRA1-missense* evolution experiments provide a benefit of approximately 200% above the parental *IRA1-missense* strain, which corresponds to approximately 350% fitness advantage per 2-Day cycle over the original ancestor strain (Fig 2E and 2F). As will be discussed later, these extremely fit mutants represent rare and complex mutations, sometimes consisting of up to 4 distinct adaptive mutations.

We also calculated the relative fitness improvement provided by auto-diploidization alone by comparing the fitness of the pure diploid population to the neutral haploids for each parental strain. Consistent with the pattern of diminishing epistasis observed from the evolution trajectories, we find that the fitness benefit of auto-diploidization has decreased in the second step of evolution from 95% per cycle in the first step of evolution to 63% (±0.6%) on average across all second-step auto-diploids (Fig 2F). However, this number varied by parental strain, with *TOR1* auto-diploids providing the largest fitness benefit of 81% per cycle and auto-diploids of Ras/PKA parental strains providing fitness benefits of 55% (±0.7%), 48% (±0.7%), and 52% (±0.8%) per cycle to *CYR1*, *GPB2*, and *IRA1-missense* strains (Fig 2F). Surprisingly, we did not isolate any auto-diploids from the *IRA1-nonsense* evolution experiments. We suspect this could be due to differences in the fitness benefit provided by auto-diploidization to *IRA1-nonsense* strains compared to other beneficial mutations in the same evolving population or reduced auto-diploidization rate in this genetic background.

## Second-step adaptive clones demonstrate a shift from pleiotropic to modular adaptation

Next, to determine whether the pattern of pleiotropic adaptation observed over the first step of evolution is maintained in the second step, we compared changes in performance for each mutant to its parental strain. To calculate each mutant's performance in fermentation, respiration, and stationary phases, we leveraged differences in each mutant's fitness in experiments of different transfer lengths, which was previously shown to be a good proxy for the direct performance in each growth phase [5,10]. In particular, a mutant's respiration performance per hour was calculated as the difference between its 2-Day fitness and 1-Day fitness, divided by the 24 h in respiration phase experienced over the second day (Fig 3A). We then used this respiration performance to extrapolate the mutant's relative fitness at 20 h, the time at which the population undergoes the diauxic shift from fermentation metabolism to respiration metabolism, with which we can calculate its fermentation performance per hour (Fig 3A). Finally, we calculated a mutant's stationary performance by taking the difference between 5- and 3-Day fitness and dividing it by the 48 h of stationary phase experienced over these 2 days (Fig 3A).

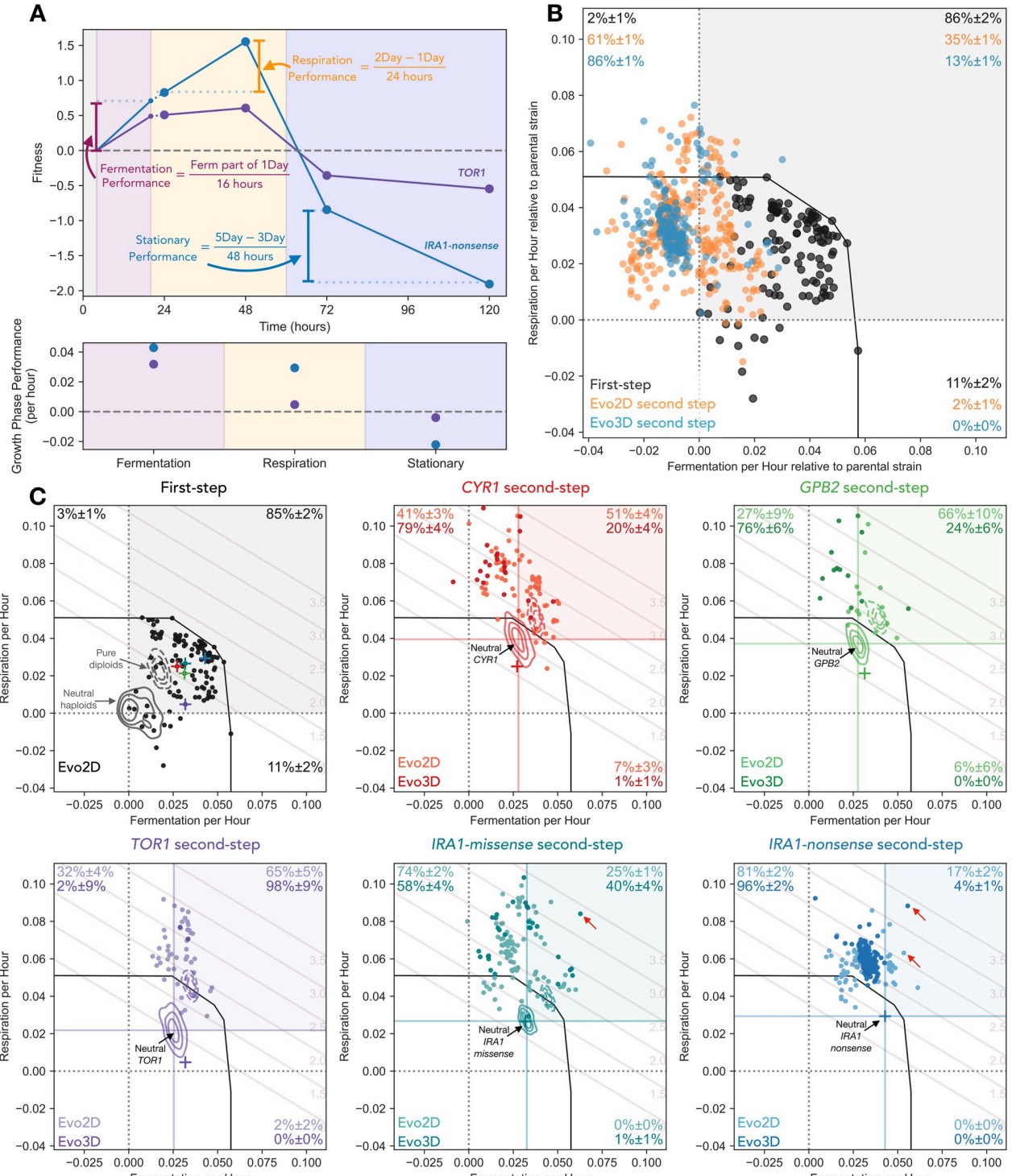

**Fig 3. Second-step adaptive mutants tend to improve respiration performance and not fermentation performance. (A)** Performance calculation in each growth phase. Respiration performance (per hour) is calculated as the difference between a mutant's 2-Day and 1-Day fitness, divided by 24 h. To calculate fermentation performance (per hour), we remove 4 h of 1-Day fitness that is due to the mutant's respiration benefit. The remaining fitness is then divided by 16 h of fermentation phase. Stationary phase performance (per hour) is calculated as the difference between 5- and 3-Day fitness divided by 48 h. Example fitnesses and performances are shown for the *TOR1* and *IRA1-nonsense* mutations used as parental strains for the second step of evolution. **(B)** Comparison of changes in performances from first- to second-step mutants relative to each parental strain. Note that first-step mutants are shown relative to the initial ancestor (the same as their measured fitness). Second-step mutants are shown relative to the relevant parental strain (i.e., second-step mutants from *IRA1*-missense are shown relative to neutral *IRA1-missense* parental lineages). Percentages in

corners indicate estimated fraction of mutants in each quadrant as determined by re-sampling of mutants with fitness measurement error. **(C)** Performance of isolated mutants separated by parental strain. Each mutant's performance in fermentation and respiration growth phases relative to the ancestral strain is shown, separated into subfigures by the initial ancestor for each mutant. Kernel Density Estimates represent the density of neutral haploids (solid lines) and pure diploids (dashed lines) for each ancestor. Crosses represent the barcoded mutants carrying the first-step mutation from the initial evolution experiment. Black line depicts a convex hull of the most extreme first-step mutants. Fitness isoclines show the 2-Day fitness advantage per cycle relative to ancestral strain associated with each location in the performance space. Red arrows indicate very fit clones that improve both fermentation and respiration which have acquired 3 or 4 adaptive mutations. The data and code underlying this figure can be found in https://zenodo.org/records/13336585.

Importantly, the growth phase performances calculated here reflect compound measurements of several parameters important to fitness during and between growth phases, including energy metabolism, sensing of changing nutrient gradients, and survival.

We found that while 85% (±3% by resampling with measurement error; see Methods) of isolated first-step adaptive mutants improved performance in both fermentation and respiration phases (black points within gray square in Fig 3B), only 35% (±1%) ($p < 0.001$ for percentage lower than first step according to a re-sampling test; see Methods) of isolated second-step adaptive haploids evolved in the same 2-Day transfer environment improved performance over their first-step parental strain in both phases (light orange points within gray square in Fig 3B). Second-step mutants that were isolated from Evo3D, which encompasses the growth phases of Evo2D, show an even stronger shift from adaptive pleiotropy than the second-step mutants from the Evo2D, with only 13% (±1%) of these mutants improving performances in both fermentation and respiration (darker colored points labeled "Evo3D" in Fig 3B and 3C). This shift is also seen for each parental strain individually (Fig 3C), with Evo2D second-step mutants isolated from each first-step parental strain showing a reduction in the number of mutations that improve performance in both fermentation and respiration, albeit with some variability in magnitude. For example, only 25% (±2%) and 17% (±2%) of second-step Evo2D mutants from *IRA1-missense* and *IRA1-nonsense* parental strains, respectively, improved both fermentation and respiration performances (Fig 3C). At the same time, 51% (±4%), 65% (±10%), and 65% (±5%) of second-step Evo2D mutations improve both fermentation and respiration performances from *CYR1*, *GPB2*, and *TOR1*. Thus while the second-step adaptive mutations are still capable of improving fermentation and respiration performances at the same time, the probability of mutations being pleiotropically adaptive is lower.

In addition to a reduction in the number of second-step mutations that improve performance in both fermentation and respiration phases, we noticed that second-step mutants were much more likely to improve respiration performance than fermentation performance. Across all second-step Evo2D mutants, 98% (±1%) improved respiration performance; 61% (±1%) of these mutants improved respiration at the cost to fermentation performance (Fig 3B). This effect is even stronger for Evo3D mutants, where 86% (±1%) improve respiration at a cost to fermentation performance (Fig 3B). This trend holds across most parental strains, with the strongest pattern seen for mutants evolved from the *IRA1-nonsense* parental strain, where 81% (±2%) of Evo2D mutants and 96% (±2%) of Evo3D mutants improved respiration performance at the cost of fermentation performance. Note that while many of these mutants reduce fermentation performance from the initial first-step parental strains, no isolated mutants had fermentation performances that were significantly worse than the original ancestor strain (Fig 3C, vertical black dashed line in each subplot).

At the same time, improving performance in only the fermentation phase is rare. Only 2% of second-step Evo2D mutants and no isolated second-step Evo3D mutants improve fermentation alone, despite the fact that an equivalent improvement in fermentation performance

would result in similarly high fitnesses for those mutants in the 2-Day condition these populations were evolved in (see fitness isoclines in Fig 3C).

Despite these general patterns revealing a shift from pleiotropic to modular adaptation, there are several examples of very strongly adaptive clones which improve both performances. For example, one clone isolated from the *IRA1*-missense population has a fitness advantage of 340% per cycle relative to the initial ancestor (or 210% relative to the *IRA1*-missense parental strain). This mutant does improve performance in both fermentation and respiration growth phases, albeit with most of its fitness gain coming from respiration (Fig 3C, *IRA1*-missense panel, labeled with red arrow). We isolated other rare examples of very fit clones that improve both growth phases from other parental strains (Fig 3C, *IRA1*-nonsense panel, labeled with red arrows), suggesting that the yeast have not yet reached functional constraints on the ability to improve both fermentation and respiration performance and that it is still possible to improve both performances beyond the evolutionary constraints observed for the first step of adaptation. As discussed below, some of these very fit clones have acquired third or fourth adaptive steps, allowing them to achieve these high fitnesses.

## Adaptively modular second-step mutants are more likely to improve performance in stationary phase

We next asked how the shift from adaptive pleiotropy to adaptive modularity of performances under selection affects how these mutants perform in other tasks not under selection in the Evo2D environment. For example, Li and colleagues [10] showed that many of the first-step mutations, which tended to improve both fermentation and respiration performances, exhibited costs in stationary phase performance. Does the shift towards adaptive modularity reduce the likelihood or magnitude of costs in other performances, potentially indicating that these mutants are more modular overall? Or do these costs to other performances remain?

To address these questions, we calculated each mutant's performance in stationary phase (Fig 2A). As previously described [10], first-step mutants are more likely to incur a cost in stationary phase than to improve it (Fig 4A), with 30% (36/119) of mutants showing such a cost (Fig 4B) and less than 2% (2/119) showing any improvement in stationary performance. The most fit first-step mutants which improve both fermentation and respiration performance to substantial degrees tend to have larger costs in the stationary phase. In particular, the *IRA1*-nonsense mutants, which were the most fit in the first-step, have the strongest costs in stationary phase performance, up to −4% per hour (Fig 4A).

We find that many Evo2D second-step mutants do pay a cost in stationary phase. In particular, 42% of second-step Evo2D adaptive mutants have significantly lower stationary performance than their parental strain. At the same time, these costs to stationary performance tend to be somewhat minor, exhibiting costs of less than −2% per hour for second-step mutants derived from *CYR1*, *GPB2*, *TOR1*, or *IRA1*-missense parental strains (Fig 4B). Note that this was not the case for the second-step mutants isolated from *IRA1*-nonsense populations, which exhibited the strongest costs to stationary phase in the first step of evolution. These second-step mutants had further costs to stationary performance as extreme as −10% per hour (see S4 and S5 Figs).

However, in contrast to the first step of evolution where stationary performance was rarely improved, 25% (77/306) of second-step Evo2D mutants show an increase in stationary performance (Fig 4A). We further stratified the second-step mutants based on their combined fermentation and respiration performances. Specifically, we asked whether mutants that only improve respiration performance showed behavior in stationary phase that was distinct from those that improved both fermentation and respiration performance. We find that second-

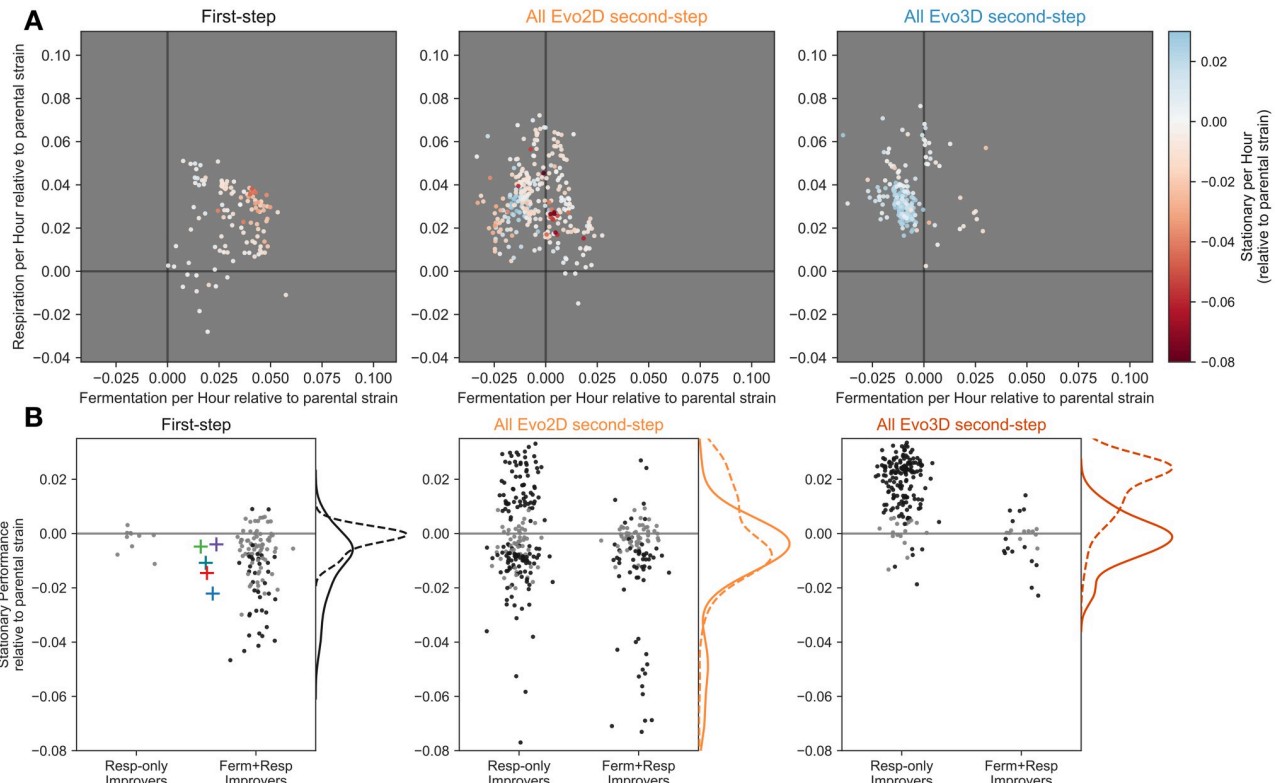

**Fig 4. Mutations that improve respiration performance only exhibit less extreme costs in stationary phase compared to those that improve fermentation and respiration performances. (A)** Each panel depicts mutant performance relative to their parental strain, colored by the relative stationary phase performance of the parental strain (see color bar). The first panel shows all first-step mutants. The second and third panels depict second-step mutants isolated from Evo2D and Evo3D conditions, respectively. **(B)** Each panel shows the stationary performance relative to the mutants' respective parental strains. Mutants are split according to the effect on fermentation and respiration performances. Those which improve both fermentation and respiration are categorized as "Ferm+Resp Improvers" and all other mutants are categorized as "Resp-only improvers." Black points represent those with measurement error that does not overlap 0. Gray points have measurement error that shows no significant change in stationary performance relative to the parental strain. Panels are organized as in (A). Kernel density estimates show the relative density for respiration-only improvers (dashed line) and fermentation and respiration-improving mutants (solid line). The data and code underlying this figure can be found in https://zenodo.org/records/13336585.

step Evo2D mutants that only improved respiration performance had varied effects on stationary performance, with 36% (70/197) showing increased stationary performance and 39% (76/197) showing a cost to stationary performance. By contrast, second-step Evo2D mutants that improved both fermentation and respiration performances were much less likely to improve stationary performance, with only 6% (7/109; $p < $ 1e-8 compared to by respiration-only improvers by Fisher's exact test) of these mutants showing stationary improvement and 49% (53/109) imposing a cost on yeast's ability to survive stationary phase.

Thus, it appears that mutations that are capable of improving both fermentation and respiration at the same time are more likely to incur costs in stationary phase. This inherent relationship may explain the reduction in Evo3D mutants that improve both fermentation and respiration performances (Figs 3B and 4A), as stationary phase is additionally under selection in this condition. Indeed, 79% (160/202; $p < $ 1e-8 compared to Evo2D by Fisher's exact test) of Evo3D mutants show an improvement in stationary phase, 97% (155/160) of which do not improve fermentation. While 7% (15/202) of Evo3D mutants do exhibit a cost in stationary phase, these costs are relatively minor and are primarily found in mutants with combined

fermentation and respiration performances that compensate for these costs to stationary performance (Fig 4A and 4B). These data indicate that the shift from mutants that improve both fermentation and respiration performances to those that primarily improve respiration performance is accompanied by a change in stationary phase performance. This pattern is true even when the other performance (stationary phase) is not under selection, as is in the case of Evo2D, suggesting that the pleiotropic "side effects"—that is phenotypic effects of mutations that are not primarily under selection [20]—of these second-step mutants may differ more generally from those of the first-step mutants.

## Changes in selection pressure and physiological limitations do not explain the shift towards modular adaptation

Thus far, we have shown that there is a general shift in the effect that adaptive mutations have on performance in growth phases over the course of a two-step adaptive walk. In particular, we find that while first-step mutations exhibit adaptive pleiotropy, improving both fermentation and respiration performances, second-step mutations isolated from the same Evo2D environment tend to be adaptively modular, improving only respiration performance and often at the cost to fermentation performance. What could be driving this shift? There are 3 primary possibilities for this observation. One possibility is that, while care was taken to ensure the evolution condition was as consistent as possible to the first step of evolution, the selection pressure in the second-step evolution experiments was shifted to favor respiration performance more than fermentation performance. A second possibility is that the populations have reached physiological limits on the yeast's ability to ferment glucose, such that there is more room to improve respiration performance. Finally, it could be that genetic and signaling pathways are wired such that there are only a limited number of mutational targets available to further improve both fermentation and respiration performances in the second-step of evolution.

The first possible explanation for the shift towards modular adaptation is that the second-step of evolution was accompanied by a change in the relative contribution of fermentation and respiration growth phases to fitness in the 2-Day transfer condition. While we took care to ensure that the population sizes, transfer times, media conditions, and other details were identical to the conditions used in the first-step evolution experiment, it could be possible that differences remain. For example, the identity of the strain comprising the majority of the population in second-step evolution experiments may have shifted the selection pressures to increase the importance of respiration performance compared to the first step of evolution. To test whether there was such a shift, we compared the fitness effects of mutations in the evolution experiment itself with our fitness measurement experiments, which more closely mimic the first-step evolution experiments because the ancestral strain comprises the majority of the population. Fitness from both methods of estimation are correlated (Pearson correlation 0.7), suggesting that there is not a systematic difference in selection pressure. We explicitly looked for a systematic difference in the contribution of respiration performance to fitness by conducting a partial correlation analysis. If respiration performance contributes more to evolution fitness than expected from our fitness measurement experiments, we would expect a positive partial correlation after this accounting for fitness. However, this is not the case (r = −0.02, $p$ = 0.74), indicating that the shift from pleiotropic to modular adaptation is not due to a change in selection pressure (see Methods, Differences in selection pressure do not drive shift towards modular adaptation).

The second possible explanation for the shift towards modular adaptation is that the yeast have reached physiological limits on the ability to improve fermentation performance. To test whether yeast have reached the upper limits of fermentation performance, we performed

additional evolution experiments in a 1-Day transfer environment, which primarily selects for fermentation performance. From these experiments, we isolated at least 1 second-step mutation from the *IRA1*-nonsense population with fermentation performance of 6.7% (±0.19%) per hour, which significantly exceeds the highest fermentation performances achieved by first- or second-step mutations evolved in the 2-Day and 3-Day environments (5.7% per hour; see S2 Fig). This suggests that a physiological maximum in fermentation performance has not yet been reached after the first adaptive step.

Having excluded a change in selection pressure and physiological constraint as explanations for the shift towards modular adaptation, we turn to the third possibility: that cellular wiring limits the simultaneous improvement of both fermentation and respiration performances by a single mutation. In particular, the preexisting wiring of genetic and signaling pathways may be such that, after acquiring a first mutation that improves both fermentation and respiration, it is much easier to find mutations that improve respiration performance at the cost of fermentation performance than it is to find mutations that improve both performances or even fermentation performance at the cost to respiration performance.

### Second-step adaptive mutations reveal a shift from mutational targets in general nutrient-sensing pathways to specific processes involved in mitochondrial function

To better understand these patterns of pleiotropy and to identify the genetic basis of adaptation in these environments, we performed whole genome sequencing on 324 adaptive mutants and identified variants (see Methods). To identify the likely adaptive mutations, we compared the genes across all isolated mutants from all evolution experiments and labeled genes that were hit more than 3 times across all mutants as putatively causal, removing mutations that were represented across all isolated mutants derived from a given parental strain, likely reflecting mutations that arose during the barcoding process. After identifying genes that were recurrently targeted, we further identified genes belonging to the same pathways and labeled these as putatively causal as well.

From this whole genome sequencing, we found some adaptive targets in nutrient-sensing pathways that were previously identified in the first step of evolution. The first step of adaptation typically involved mutations in one of 2 signaling pathways responsible for sensing glucose and instructing the cells to grow: the Ras/PKA and TOR/Sch9 pathway [9] (Fig 5). Most of these mutations resulted in loss of function in negative regulators of the pathway or modification of function in positive regulators, ultimately driving constitutive activation of these pathways [9]. In an analysis of the second step of evolution for *TOR1*, *CYR1*, and *GPB2* mutants in the 2-Day environment, Aggeli and colleagues [19] identified Ras/PKA pathway mutations as an adaptive route for *TOR1* mutants and TOR/Sch9 mutants as an adaptive route for *CYR1* and *GPB2* mutants. The additional sampling we have conducted here, including sequencing mutants isolated from the 2 *IRA1* populations under Evo2D and Evo3D, further confirm that TOR/Sch9 pathway mutations are commonly observed in the background of Ras/PKA mutants. In particular, we find that mutations in the gene *KSP1*, a PKA-activated kinase which inhibits autophagy via TORC1 [27,28], are common across all of the Ras/PKA parental strains (Fig 5). These mutations were most commonly isolated from *IRA1*-nonsense populations, where 42% (32/77) of Evo2D mutants and 91% (30/33) of Evo3D mutants harbored a *KSP1* mutant. Unlike the TOR/Sch9 pathway mutants observed in the first step of evolution, which putatively result in increased TORC1 activity, increased cell growth, and decreased autophagy [9,29], many of the observed second-step *KSP1* mutations are loss-of-function. This indicates that these mutations may be acting in the opposite direction of first-step mutations

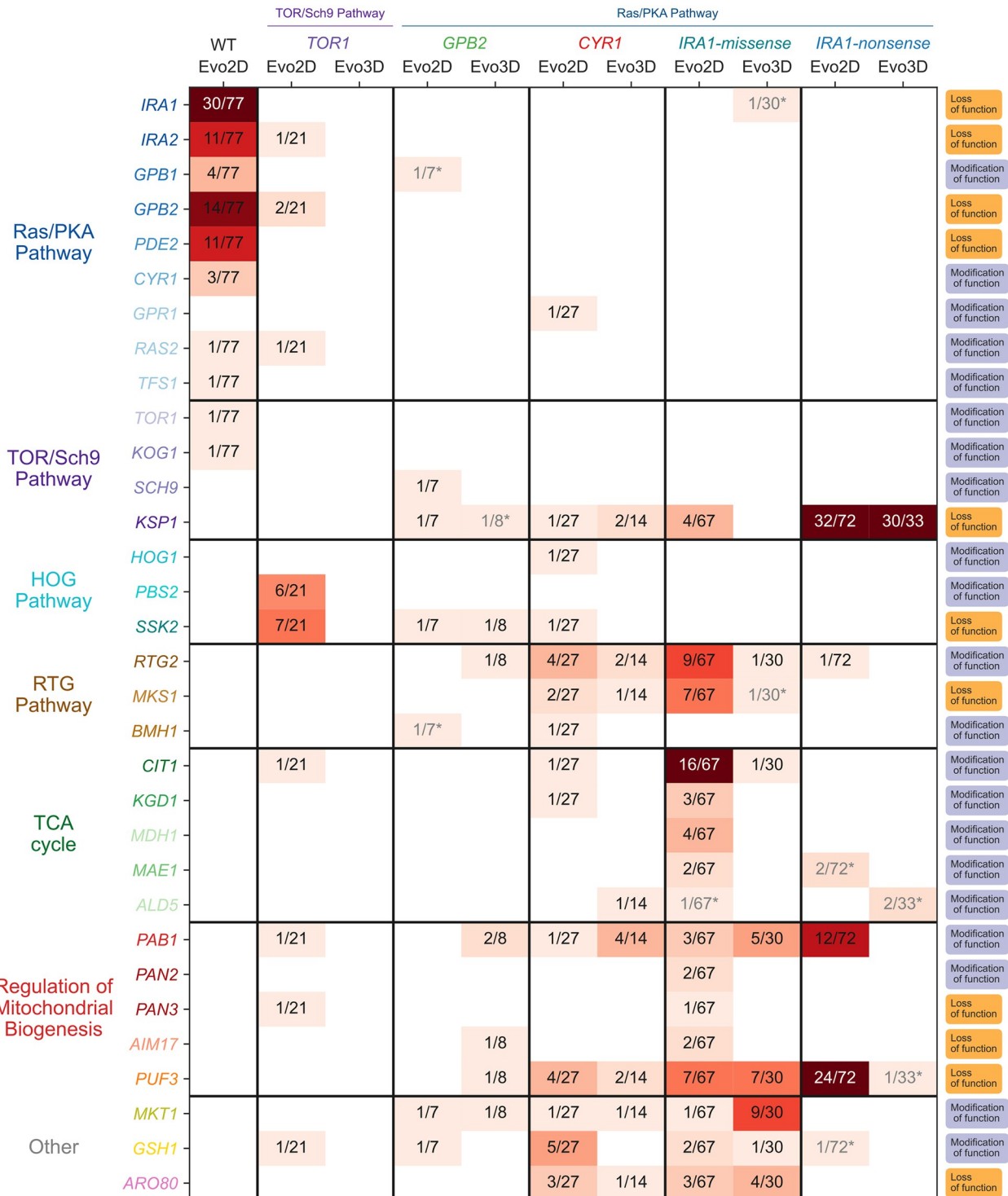

**Fig 5. Identified mutations by ancestral genotype and evolution condition.** The figure depicts the number of putative adaptive mutations identified in each gene, split by parental strain and evolution condition. The denominator in each entry corresponds to the total number of putative adaptive mutations identified in the corresponding parental strain and evolution condition. Boxes with gray text and asterisks indicate genes mutated only in the context of other putatively causal mutants. The column on the far right indicates the putative functional effect of the mutations on the gene. If any stop-gained or frameshift mutations were identified in this gene, it was classified as harboring "loss of function" mutations. If instead, only missense or nearby non-genic mutations were identified, the gene is classified as "modification of function."

in this pathway, potentially allowing for the up-regulation of TOR despite (or in compensation of) increased activation of PKA associated with the Ras/PKA mutants.

Beyond mutations in nutrient-sensing signaling pathways commonly being observed in the first-step of adaptation, our sampling reveals a shift towards mutational targets related to mitochondrial function and respiration, which likely affect the respiration performance of mutants measured in our study (Fig 5). In particular, we find that 36% (22/64) of adaptive mutants isolated from *IRA1*-missense populations in the 2-Day evolution condition acquire mutations in or near genes involved in the TCA cycle (*CIT1*, *KGD1*, *MDH1*, *MAE1*, *ALD5*). Interestingly, all of these mutations are either missense or putatively regulatory mutations in enzymes directly responsible for respiration, suggesting they may modify the function or expression of these enzymes, potentially changing respiratory flux [30–37]. In addition, we identified mutations in several genes related to the regulation of respiration and mitochondrial function, with 25% (16/64) of isolated 2-Day *IRA1*-missense mutants identified as carrying a mutation in the RTG pathway, which is responsible for the regulation of genes important for respiration. In particular, we observe putative loss-of-function mutations in *MKS1*, a negative regulator of the RTG pathway, and missense mutations in *RTG2*, a positive regulator of the pathway [38–42]. This suggests that these mutations may be up-regulating the RTG pathway and the genes it regulates, indirectly increasing metabolic flux through the TCA cycle. Moreover, 19% (12/64) of these mutants carry a mutation in other genes related to the regulation of mitochondrial biogenesis (*PUF3*, *PAB1*, *PAN1*, *PAN2*, *AIM17*), many of which are related to posttranscriptional modification of mRNA molecules related to mitochondrial function or respiration [43–45].

These patterns are also observed in other populations harboring different first-step mutants. In particular, while our sampling for *IRA1-nonsense* and *IRA1-missense* populations allowed us to detect the largest number of mutational targets, mutations in genes involved in the TCA cycle, RTG pathway, and mitochondrial biogenesis were found in populations from nearly all first-step mutations, with a few exceptions. These exceptions, for example the absence of HOG-pathway mutations from *IRA1*-missense and *IRA1*-nonsense backgrounds, is suggestive of historical contingency, where the identity of further mutations is dependent on mutations acquired earlier in evolution [46–49]. While some of these genes were detected in previous work [19], additional sampling from new evolution experiments in the 3-Day condition and additional parental strains (*IRA1*-missense and *IRA1*-nonsense) allowed us to more confidently identify recurrently mutated genes and to group the observed sets of mutations and genes into functional categories and pathways.

## Mutations isolated from the 3-Day evolution experiment are a subset of 2-Day adaptive mutants

We next examined whether there was a difference in the mutational targets isolated from 2- and 3-Day evolution experiments, given that we observed that Evo2D mutants were more likely to improve both fermentation and respiration performances than Evo3D mutants (Fig 3), and Evo3D mutants were more likely to improve stationary performance than Evo2D (Fig 4). In particular, we wondered whether the addition of stationary phase as a selective pressure allowed for new mutational targets to be adaptive because of their effect on stationary phase, or if instead, the addition of stationary phase restricted the Evo3D mutational targets to a subset of the Evo2D mutations that were not costly to stationary performance.

By comparing the sets of mutated genes for well-sampled parental strains *IRA1-missense* and *IRA1-nonsense*, we saw that all genes mutated in the 3-Day evolution experiments were also identified in the 2-Day evolution experiments (see Fig 5). In particular, *PUF3*, *PAB1*, and

*MTH1* mutants are entirely absent as single mutations from the 3-Day *IRA1*-nonsense experiments, shifting the molecular targets to essentially just those in *KSP1*. Similarly, RTG and TCA cycle mutants are reduced in frequency or absent from the 3-Day *IRA1*-missense experiments, respectively. As expected, these mutations that are reduced in frequency show costs in stationary performance and thus have reduced fitness in the 3-Day transfer environment (S9 and S10 Figs).

## The reduction of mutational targets in the Ras/PKA pathway drives the shift towards modular adaptation

To understand how the shift from pleiotropic to modular adaptation over the two-step adaptive walk is reflected on a molecular basis, we examined how each of these mutations moved the organisms in the performance space. The first step of evolution, which primarily hit mutational targets in the Ras/PKA pathway, shows strong patterns of pleiotropic adaptation, with these mutations improving both fermentation and respiration performances (Fig 6A).

Of the second-step adaptive haploids, those with mutations in the Ras/PKA pathway (Fig 6B, blue circles), which were isolated primarily from the *TOR1* populations, also display pleiotropic adaptation, improving both fermentation and respiration performance. This suggests that mutations which putatively increase the activity of the Ras/PKA pathway are indeed generally adaptively pleiotropic.

In addition to Ras/PKA mutations, other haploids with mutations in *ARO80* (Fig 6B, pink circles) and *GSH1* (Fig 6B, gold circles) show recurrent patterns of pleiotropic adaptation across parental strains, notably across *CYR1* and *IRA1*-missense genetic backgrounds (S9 and S10 Figs). Mutations in these genes, which are involved in amino acid catabolism [50,51] and glutathione biosynthesis [52], respectively, may be adaptively pleiotropic due to their involvement in processes entirely orthogonal to, or upstream of, both fermentation and respiration.

Many of the remaining mutational targets improve respiration performance at the cost of fermentation performance. In particular, haploid mutants which harbor mutations in genes involved in the TCA cycle (Fig 6B, green circles), mitochondrial biogenesis (orange, red circles), or the RTG pathway (brown circles) improve respiration performance at the cost to fermentation performance in *CYR1*, *GPB2*, *TOR1*, and *IRA1*-missense backgrounds when present (Fig 6). Notably, haploids that harbor mutations in these genes have similar fitness in the 2-Day transfer environment to mutants with mutations in *ARO80* and *GSH1*, which exhibit adaptive pleiotropy. Despite these similar fitnesses, there is an 8-fold increase in observed adaptively modular genetic targets than those that are adaptively pleiotropic in the *IRA1*-missense 2-Day evolution experiments (41 mutants in TCA and RTG with fitnesses between 2.0 and 2.5 compared to 5 in *GSH1* and *ARO80* for *IRA1*-missense).

There are also single point mutations in *MKT1* which achieve very high 2-Day fitness by greatly improving respiration performance and showing little cost to fermentation performance (Fig 6B, chartreuse circles). Interestingly, all adaptive mutations in this gene occur at the same nucleotide, changing from 89A to C, G, or T. Thus, while these mutations are driven by only a single mutation, their lower frequency reflects the reduced target size compared to the other haploid mutations which have multiple targets within the gene (e.g., those in RTG pathway, TCA cycle). The 89A allele is a derived allele in the parental S288C yeast strain used for all of these experiments and reflects an ancestral reversion in the case of A89G. This A89G reversion has been previously observed in other evolution experiments in glucose limitation and the 89G allele has been shown to stabilize mRNA of mitochondrial genes that are targets of Puf3 [43,53]. Interestingly, 89C and 89T alleles each provide similar fitness benefits as the

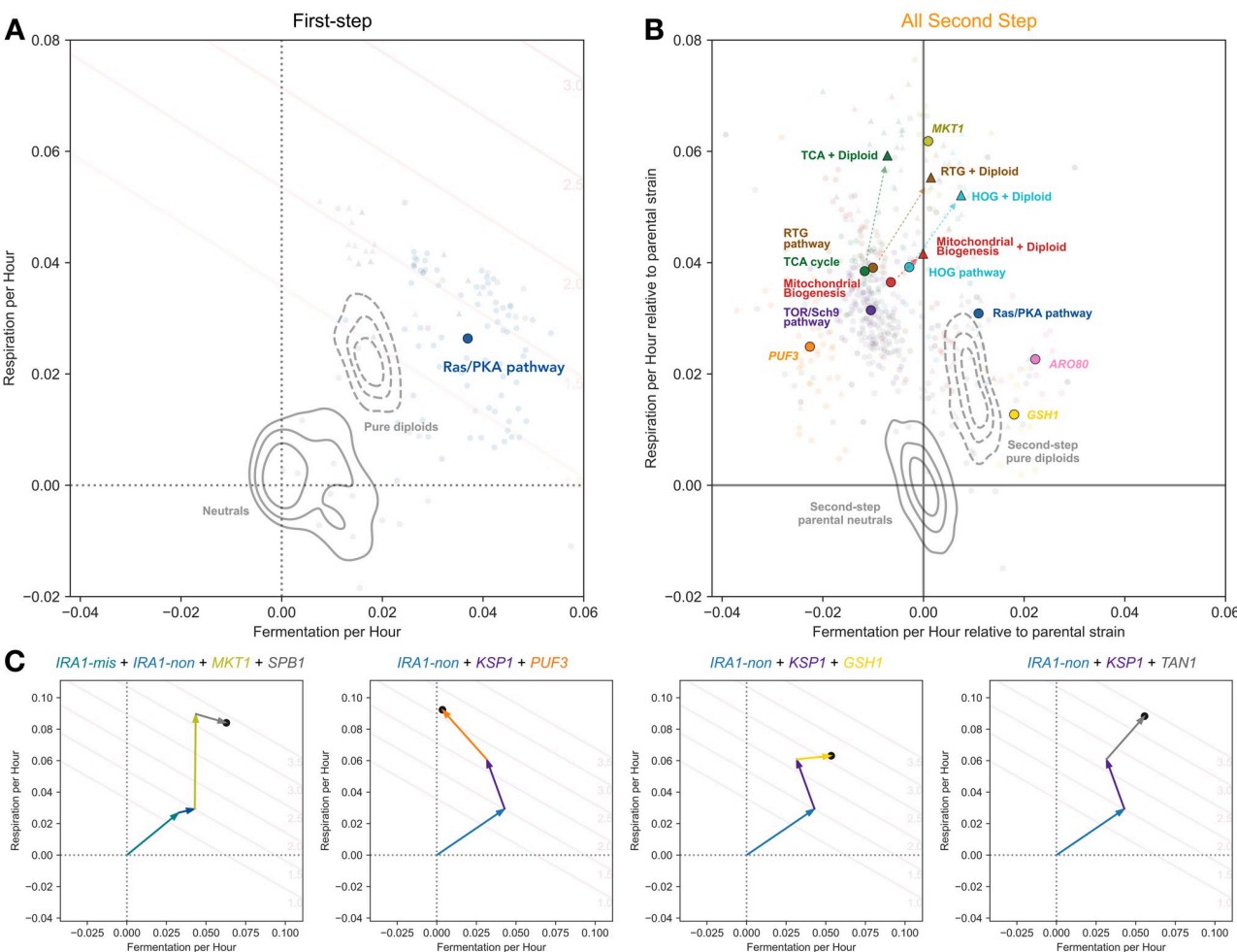

**Fig 6. Adaptive modularity is driven by the accessibility of mutational targets that improve respiration at the cost of fermentation. (A)** Ras/PKA and TOR/Sch9 mutants from the first step of adaptation to improve both fermentation and respiration performance. **(B)** Most common second-step mutational targets tend to improve respiration at the cost of fermentation (centroids depicted as colored circles), except for rare Ras/PKA (blue), *ARO80* (pink), or *GSH1* (yellow) mutants; haploids shown as circles. Auto-diploids exhibit adaptive pleiotropy (dashed Kernel Density Estimates for all parental strains, colored by first-step mutation). Auto-diploidization is also adaptively pleiotropic on the background of other point mutations (triangles colored by pathway or gene category). Note that only centroids for each category of gene with at least 3 observed mutants were included. **(C)** Triple and quadruple mutants can ultimately drive adaptation towards adaptive pleiotropy (or adaptive modularity) despite being primarily composed of adaptively modular mutations. Note that the mutations beyond the first step are depicted in no particular order in these subpanels. The data and code underlying this figure can be found in https://zenodo.org/records/13336585.

89G allele in our experiments despite resulting in distinct amino acids, suggesting that the 89A allele and the resulting aspartic acid may be particularly costly to *MKT1* function.

Beyond adaptive mutations in haploids, auto-diploidization is a common mode of adaptation. In particular, we see that in addition to diploidy being adaptively pleiotropic by itself (pure diploids shown as topographical Kernel Density Estimates in Fig 6), high-fitness diploids that co-occur with other mutations (colored triangles in Fig 6) also show patterns of adaptive pleiotropy, improving both fermentation and respiration performances. This seemingly universal benefit without cost likely explains the high frequency of auto-diploidization observed across genetic backgrounds and environmental conditions in many yeast evolution experiments [7,9,54–56]. While these high-fitness diploids provide a much larger benefit than haploids that harbor mutations in the same genes, their reduced frequency is likely due to a lower

mutation rate, as these mutants needed to acquire both a mutation in an adaptive target and auto-diploidize, together improving respiration to a larger extent and mostly eliminating costs to fermentation performance associated with the mutation. Notably, most of the point mutations are homozygous, indicating they likely occurred before the auto-diploidization event.

In addition to these general trends, we sampled a small number of mutations that have a total of 3 or 4 putatively causal mutations since the original ancestor. These mutants provide hints about how adaptation might proceed over longer adaptive walks. In one case, as demonstrated by the *IRA1*-nonsense + *KSP1* + *PUF3* mutant depicted in the second panel of Fig 6C, we observe adaptation as continuing down a route towards specialization in respiration performance. We also observe 3 examples where the collective effect of the mutations instead drives evolution towards generalism—improving both fermentation and respiration performance— despite being composed of second-, third-, and fourth-step mutations which tend to improve only 1 performance or the other. For example, one *IRA1*-missense mutant acquired an *IRA1*-nonsense mutation, an *MKT1* A89G mutation which improves only respiration by itself in this background, and acquired a mutation in *SPB1* (Suppressor of *PAB1*), which is expected to improve only fermentation with a modest cost to respiration, assuming additive mutational effects in the performance space (Fig 6C, first panel). We see similar examples for 2 *IRA1*-nonsense mutants: one of which acquired both *KSP1* and *GSH1* mutations and the other of which acquired *KSP1* and *TAN1* mutations (Fig 6C, third and fourth panels), where the collective effects of the observed mutations ultimately continue to push the population towards improving both traits. These rare mutants demonstrate that, at least on short evolutionary timescales, navigation of the performance space seems to be more driven by constraints imposed by the genetic wiring of the cell, which influences the relative ease of improving one performance or the other, rather than fundamental or physiological constraints upon improving the performances themselves.

## Discussion

In this study, we sought to understand the observation that single adaptive mutations observed in experimental evolution, especially those of large effect, can pleiotropically improve multiple distinct performances at once. This observation is puzzling because theoretical work suggests that as pleiotropy increases, large effect adaptive mutations should become less probable [1]. This expected "cost of complexity" is the reason for why modularity is often seen as the necessary condition and the expected consequence of evolution of complex organisms by natural selection [1,2,14,57–60].

We focused on one striking example of pleiotropic adaptation that comes from previous studies of yeast evolving in a glucose-limited environment [7] in which 85% of the first-step single adaptive mutations improved performance in both fermentation and respiration growth phases (Figs 1A and 3B) [7,9,10], despite these growth phases as being thought to be physiologically distinct.

Here, we investigated whether adaptation in the same low-glucose environment and 2-Day transfer as the original experiment [7] will continue following the path of adaptive pleiotropy (Fig 1B) or will shift to become more modular (Fig 1C). We thus further evolved 5 different first-step mutants, 4 in the Ras/PKA pathway (*IRA1-nonsense*, *IRA1-missense*, *CYR1*, and *GPB2*), and 1 in the Tor/Sch9 pathway (*TOR1*), sampled a large number of adaptive mutants, and evaluated their effects on the fermentation and respiration performances.

In all 5 cases, the results were qualitatively similar. First, adaptation proceeded to improve fitness, albeit to a somewhat muted degree. Second, while a number of mutants were adaptively pleiotropic, improving both fermentation and respiration performances, the dominant trend

switched towards more modular adaptation. Specifically, nearly all adaptive mutants improved respiration performance sharply and many had no or only weakly positive or even negative effects on the fermentation performance (Fig 3B). These results support a model of adaptation wherein early adaptation is typically driven by mutations of large effect that improve multiple performances at once. Then, after these mutations have become exhausted, adaptation may proceed via more modest mutations that improve performances in a stepwise manner (Fig 1C).

One remaining question is how pleiotropic adaptation is possible in the first place. The prevalence of the pleiotropic adaptation in the first step may be due to these mutations being primarily in Ras/PKA pathway genes. We thus hypothesized that the adaptive pleiotropy is a consequence of the way this pathway has evolved to shift rates of metabolism in both fermentation and respiration in a substantial, coordinated, and beneficial fashion [29]. The notion is that even though these metabolic functions are distinct, they are often required to be carried out in tandem, as respiration commonly follows fermentation for yeast. It is possible that sensing and signaling pathways such as the Ras/PKA pathway evolved to affect them together. This might be a general feature of signaling pathways as they must shift multiple functions and performances together and this ability then represents an attractive target for adaptive genetic changes.

If pleiotropic adaptation is a feature of the Ras/PKA pathway, the prediction is that second step adaptive Ras/PKA pathway mutations will remain adaptively pleiotropic. This is indeed the case. Second-step mutations in the Ras/PKA pathway, mainly arising in the *TOR1* background, do improve both fermentation and respiration performances. A small number of second-step adaptive mutations outside of this pathway, in *ARO80* and *GSH1*, are also pleiotropically adaptive and improve both respiration and fermentation, but to a smaller degree than Ras/PKA pathway mutants. This suggests that the Ras/PKA pathway is virtually unique in its ability to modulate both fermentation and respiration performances together to a substantial degree in an adaptive manner, a notion also supported by the fact that we observe the shift towards modularity adaptation already in the second adaptive step.

A small number of adaptive clones in the second step improved both fermentation and respiration performances to a substantial degree. Sequencing of these clones showed that they acquired multiple mutations, and several of these clones improved both performances by the addition of 2 or more orthogonal steps. This suggests that adaptation can continue improving both performances but the adaptive walk needs to engage multiple modules and multiple mutations, making such adaptation slower than the first step of adaptation. This might be part of the reason why adaptation in general slows down over the course of evolution [6,8,19,23].

We argue that signaling pathways such as Ras/PKA have the capacity of generating "coherent pleiotropy," where the output of many cellular processes can be affected without disrupting the proper regulation and function of each process. As such, signaling pathways that have been evolutionarily pre-wired to control combinations of selective pressures may be easily modified by mutation to coherently improve the performances under selection. The ability of signaling pathways to generate coherent pleiotropy implies that many adaptive mutations should hit signaling pathways. Indeed, this is what we see. For example, in cancer, the key oncogenes are located along cellular signaling pathways and engage either receptors of signals or represent key relay stations in these pathways [13,61–65].

It may even be that the mutations we identify here as "modular," improving respiration performance, are themselves examples of adaptive and coherent pleiotropy. Such mutations may improve a suite of traits within the measured respiration fitness component that we are unable to detect with our approach. We do observe mutations in genes involved in retrograde signaling and regulation of mitochondrial biogenesis, suggesting these mutations may result in coordinated improvements of several traits downstream of these processes.

On the other hand, this coherent pleiotropy of signaling pathways does not necessarily indicate that such mutations have no costs in other traits. Indeed, we see that many of the Ras/ PKA mutants exhibit costs in stationary phase. Moreover, in previous work, we find that the Ras/PKA mutants have additional phenotypic effects with minor contributions to fitness in the Evo2D evolution condition but substantial effects in other conditions [20]. Thus, we might expect these signaling pathways to be most likely to be targeted by adaptation in relatively simple environments where the beneficial pleiotropic effects can dominate the overall fitness effect with only minor other costs being relevant to the specific condition.

The extent to which pleiotropic adaptation drives early evolution in other systems remains to be determined. A recent meta-analysis of evolution experiments in *E. coli* found that single nucleotide variants (but not indels or structural variants) in genes with many interactions do tend to provide larger fitness benefits than those with fewer protein–protein interactions [4]. In the *E. coli* long-term evolution experiment (LTEE), *spoT* mutations arise early in several replicate populations and have been shown to both improve growth rate and ability to exit lag phase [66]. However, there are other mutations (e.g., in *hslU* and *pykF*) that arise early across parallel LTEE replicates without clear links to improving multiple traits [67]. We hope that our study motivates further investigation into the prevalence of pleiotropic adaptation and its underlying mechanistic basis in other systems.

In addition to cellular signaling pathways, other gene-regulatory, hormonal, and neuronal systems allow for organisms to be phenotypically plastic and involve coherent control of many traits of an organism. As such, we speculate that these systems may also be attractive targets for evolutionary change, as they can serve as high-leverage routes for altering many traits simultaneously. The evolution of phenotypic plasticity may pave the way for subsequent large-effect evolutionary shifts in local adaptation.

Finally, we believe that the existence of these high-leverage pleiotropic routes of adaptation must be incorporated into our thinking of the evolution of complex systems. Indeed, we commonly think of pleiotropy as purely random, with mutations shifting multiple traits at once in a random and thus largely incoherent way. This generates expectations that pleiotropy should be costly, as such incoherent shifts lead to a generically disorganized state. Given that actual organisms have low-dimensional but pleiotropic signaling and regulatory systems [2], pleiotropy can often be coherent and thus might often enhance adaptive potential and allow for surprisingly large-effect adaptive mutations. It is therefore important to think of regulation and adaptation as 2 sides of the same problem of how to change complex and tightly integrated systems in an adaptive manner.

## Methods

### Constructing barcoded populations from first-step mutants

To conduct second-step evolution experiments, we constructed barcoded populations for each of 5 mutations that arose in the original 2-Day evolution experiment [7]. Construction of the barcoded populations of *CYR1*, *GPB2*, and *TOR1* mutations was previously described [19]. To barcode *IRA1-nonsense* and *IRA1-missense* mutants, we followed a similar procedure. Specifically, we backcrossed the *IRA1* mutants (*MATα*) to GSY5375, a *MATa* ancestral S288C strain that harbors the pre-landing pad locus [19]. After sporulation and tetrad dissection, we performed Sanger sequencing to identify segregants that were *MATα*, carried the *IRA1* variant of interest, and the pre-landing pad allele at the barcode locus, ensuring the removal of the barcode initially labeling this strain. These segregants were used for downstream transformation of barcodes.

We then barcoded these strains with a low- and high-complexity barcode as described in Aggeli and colleagues [19]. We first transformed in the low-complexity barcode by PCR-amplifying a region from the L001 library, which harbors a NatMX selectable marker, half of *URA3*, an artificial intron, a low-complexity barcode sequence, and a lox66 site. We then selected for successful transformants using YPD + Nat plates and isolated 4 and 8 colonies for *IRA1-missense* and *IRA1-nonsense* strains, respectively, each with a unique low-complexity barcode. For each of these strains, we then transformed a library of high-complexity barcodes (pBAR3). After transformation, cells were grown in YP + 2% galactose for 16 h to induce Cre recombinase expression prior to selection on SC-ura plates with 2% glucose. We then estimated the number of unique transformants by counting the number of colonies grown from plating a dilution. We additionally estimated the relative number of unique transformants by amplicon Illumina sequencing using the sequencing primers described below.

To construct populations for evolution experiments, we pooled together transformants from multiple high-complexity transformations, such that each barcode was equally represented in each pool. This resulted in pools of approximately 100,000 high-complexity barcodes for each evolution experiment with the exception of Evo2D *IRA1-missense* evolution pool which contained approximately 40,000 high-complexity barcodes. Transformants were pooled such that each low-complexity barcode was only present in 1 evolution pool, allowing us later to identify evolution conditions based on the identity of the low-complexity barcode. For Evo1D experiments, a pool of *IRA1-missense* and *IRA1-nonsense* transformants was used, containing equal numbers in abundance, albeit with approximately 32,000 unique *IRA1-missense* barcodes and approximately 60,000 *IRA1-nonsense* barcodes. A single pool that contained barcoded populations of *CYR1*, *GPB2*, and *TOR1* mutants was used for the second-step Evo1D and Evo3D experiments for these genotypes.

## Conducting evolution experiments

We conducted evolution experiments with barcoded populations under identical conditions to the original evolution experiment. Briefly, ~$10^8$ cells of each evolution population pool was inoculated in 50 ml of SC-ura + 2% dextrose + hygromycin in 500 ml Delong flasks and grown overnight at 30°C with shaking at 223 rpm, and 500 μl of saturated overnight culture was then transferred to 100 ml of glucose-limited M3 medium (5× delft medium with 4% ammonium sulfate and 1.5% dextrose) in 500 ml Delong flasks for the evolution experiment. For most of the evolution experiments, the culture was split into 2 replicate flasks at this point. Second-step Evo3D experiments from *IRA1-missense* or *IRA1-nonsense* mutants used 3 replicates each. Cultures then propagated every 24, 48, or 72 h for Evo1D, Evo2D, and Evo3D conditions, respectively. At the time of transfer, a set volume was transferred into 100 ml of fresh medium. In order to keep the bottleneck size consistent at approximately $5 \times 10^7$ viable cells, the volume varied by condition. Evo2D conditions used 400 μl of transfer volume. Evo1D and Evo3D conditions used 500 μl of transfer volume, which accounted for decreased cell density and decreased cell viability in these conditions, respectively. Two 1 ml volumes of saturated culture were frozen as glycerol stocks. The remaining culture was spun down, resuspended in 5 ml of sorbitol freezing solution (0.9 M sorbitol, 100 mM Tris (pH 7.5), 100 mM EDTA) and frozen at −20°C for subsequent genomic DNA extraction and barcode library sequencing preparation.

Evo2D experiments were carried out for a total of 22 transfers (approximately 176 generations; approximately 8 generations/transfer). Evo1D experiments were carried out for 19 transfers (approximately 133 generations; approximately 7 generations/transfer). Evo3D

experiments were carried out for 21 transfers (approximately 168 generations; approximately 8 generations/transfer).

## Isolation of clones from evolution experiments

To isolate clones for fitness measurement experiments, quantification of growth phase performances, and whole genome sequencing, we sorted individual cells from the final transfer as previously described [5]. Specifically, we sorted 480 individual cells (five 96-well plates) from each replicate evolution experiment into single wells of a 96-well plate with 100 μl of YPD medium. This resulted in a total of 80 plates (approximately 7,680 sorted cells) across the 16 evolution experiments. Sorted cells were then grown at 30˚C for 3 days without shaking until the cells reached saturation. Saturated cultures (5 μl) were then transferred to deep-well 96-well plates with 300 μl of YPD. After 2 days of growth at 30˚C without shaking, 100 μl of culture were mixed with glycerol and frozen at −80˚C, and 20 μl of saturated culture were transferred to 96-well PCR plates and frozen at– 20˚C for barcode identification. Saturated culture was also plated onto Benomyl plates to assay ploidy [9].

## Barcode identification by Metagrid

To identify the barcode associated with each well and ensure that multiple clones with the same barcode were not kept for downstream fitness measurement experiments, we performed sequencing on the barcodes of the clones in each well. Saturated culture (20 μl) was transferred to 96-well plates and frozen at −20˚C. Cells were then lysed by incubation at 95˚C for 15 min, and 5 μl of lysed culture were used as the template for PCR amplification of the barcode region. We performed 2 steps of PCR. In the first step of PCR, we used a set of 72 forward and 64 reverse first-step primers, each with a unique 8-bp multiplexing tag, to combinatorially label each well. After the first step of PCR, 5 μl of each well's PCR product from 5 plates was pooled together and the appropriate 250 bp band was isolated using gel purification. A second step of PCR was then performed with standard Nextera primers. Amplicon libraries were then sequenced on Illumina MiSeq or HiSeq machines.

To computationally identify the barcodes associated with each well, we used BarcodeCounter2 to extract the multiplexing indexes and barcode regions from each read. We then associated barcodes with each well by taking the barcode with the most reads per well, provided the well had at least 200 reads, the barcode was at least 60% of the well's reads, and it received more than 1.5× the second-highest barcode in the well. This resulted in identifying the locations of 1,785 unique barcoded clones. This is lower than the highest possible number of 7,680 clones due to a combination of some wells receiving multiple clones, multiple wells receiving cells of the same barcode, and drop out due to sequencing depth. To further validate our approach, we randomly selected 3 wells per plate and performed Sanger sequencing of their barcodes. Of the wells where both barcodes were identified using the metagrid approach and Sanger reads were of sufficiently high quality, over 85% of the barcodes matched. We subsequently pooled each uniquely barcoded clone by evolution condition and parental strain, resulting in 4 pools of barcoded lineages to be used for fitness measurement experiments.

## Benomyl ploidy test

To characterize the ploidy of each sorted clone, we performed a high-throughput ploidy test that was previously developed [9]. Saturated culture from cell sorting was pinned onto YPD agar plates containing 20 mg/ml benomyl. Plates were then grown at 25˚C for 2 days and then imaged. Clones with inhibited growth on the benomyl medium were identified as diploids.

Clones with normal growth on the benomyl medium were identified as haploids. See "Mutation and ploidy classification" section below.

## Constructing barcoded pools

To construct a pool of lineages for fitness measurement experiments, we generated one large pool of barcoded lineages isolated from previous evolution experiments and the evolution experiments described in this study. Briefly, one tube of each barcode pool was thawed and grown in YPD at 30°C overnight. After the overnight growth, we pooled all barcode-sub pools together, adjusting for the number of barcodes in each pool and the OD600 of the culture, such that each barcode was equally represented in this big pool. This big pool was then split into 1 ml glycerol stock aliquots and frozen at −80°C.

To precisely measure the mean fitness of the population, we constructed 2 pools of 60 neutral lineages from Venkataram 2016 and Li 2019. Briefly, we identified barcodes that exhibited neutral fitness estimates across all previous experiments done with these pools of barcoded lineages [5,9,10,20]. We then streaked out from glycerol stocks onto YPD plates. A single colony was picked from each barcoded lineage and grown in 96-well deep-well plates for 2 days. Wells for each collection of 60 neutrals were then pooled equally by volume. Then, glycerol stocks were created with 1 ml of pooled culture and frozen at −80°C.

## Fitness measurement experiments

To quantify fitness effects, we performed fitness measurement experiments. We streaked out DPY256 (an ancestor strain which harbors an ApaLI restriction site in the barcode region) onto a YPD plate. After 2 days of growth, a colony was picked and grown up in 50 ml of YPD overnight. Additionally, one tube each of the 60-neutral pool from Venkataram 2016 and one tube of the 60-neutral pool from the Li 2019 pool was thawed and grown separately in 50 ml of YPD overnight.

The $5 \times 10^7$ cells from DPY256 ancestor, each of the 2 neutral pools, and the big pool (see Constructing barcoded pools) were then separately inoculated into four 500 ml Delong flasks containing 100 ml of M3 medium for one cycle of pre-culture in the selective condition. This resulted in a total of 16 flasks of culture, corresponding to each set of barcoded cells and the 4 conditions.

After one cycle of growth (which corresponded to 24 h for the 1-Day transfer condition, 48 h for 2-Day, 72 h for 3-Day, and 120 h for 5-Day), the cultures were pooled by volume such that the big pool of barcoded lineages represented 2% or 5% of the population. In the 2% flasks, 2% of the population was the big pool of evolved lineages, 2% were Venkataram 2016 neutrals, 2% were Li 2019 neutrals, and 94% of the population was DPY256 ancestor. In the 5% flasks, 5% of the population was the big pool of evolved lineages, 2% were Venkataram 2016 neutrals, 2% were Li 2019 neutrals, and 91% of the population was DPY256 ancestor. These pools of lineages are considered "Timepoint 0" for each condition and pooling percentage.

We then transferred a set volume of this pool to replicate flasks (2 replicates for 1- and 2-Day experiments, 3 replicates for 3- and 5-Day experiments) containing 100 ml M3 medium such that approximately $5 \times 10^7$ of viable cells were transferred. This volume was 500 μl for 1-, 3-, and 5-Day experiments and 400 μl for 2-Day experiments. The culture was then grown at 30°C in an incubator shaking at 223 RPM. After the set amount of time corresponding to each condition, a fixed volume of culture (500 μl for 1-, 3-, and 5-Day experiments and 400 μl for 2-Day experiments) to fresh 100 ml of M3 medium in 500 ml DeLong flasks. This serial dilution was continued for until transfer 6 for 1- and 2-Day experiments and until transfer 2 for 3- and 5-Day experiments.

After each transfer, the remaining culture was frozen for downstream DNA extraction, barcode amplification, and sequencing. To freeze the culture, we transferred the culture to 50 ml conical tubes, spun down at 3,000 rpm for 5 min, resuspending in 5 ml sorbitol freezing solution (0.9 M sorbitol, 0.1 M Tris-HCL (pH 7.5), 0.1 M EDTA (pH 8.0)), aliquoted into three 1.5 ml tubes, and stored at −80˚C.

## Genomic DNA extraction

Genomic DNA was extracted from frozen cells as described previously [19]. Briefly, 400 µl of frozen cells in sorbitol solution was spun down at 3,500 rpm for 3 min. After discarding the supernatant, the cell pellet was then washed in 400 µl of sterile water and spun down at 3,500 rpm for 3 min and the supernatant was discarded. The cell pellet was then re-suspended in 400 µl of extraction buffer (0.9 M sorbitol, 50 mM Na phosphate (pH 7.5), 240 µg/ml zymolase, 14 mM β-mercaptoethanol) and incubated at 37˚C for 30 min. We then added 40 µl of 0.5 M EDTA, 40 µl of 10% SDS, and 56 µl of proteinase K (Life Technologies 25530–015), vortexing after each addition. The mixture was then incubated at 65˚C for 30 min. After the incubation, tubes were placed on ice for 5 min and then 200 µl of 5 M potassium acetate were added and tubes were shaken to mix. Following a 30-min incubation on ice, the samples were spun for 10 min at 17,000 rpm. The supernatant was transferred to a new 1.5 ml tube containing 750 µl of isopropanol and placed on ice for 5 min. We then spun the samples at 17,000 rpm for 10 min and discarded the supernatant. The DNA pellet was then washed twice with 750 µl 70% ethanol, each time vortexing very briefly, spun at 17,000 rpm for 2 min, and discarding the supernatant. After allowing the DNA pellet to dry completely, it was resuspended in 50 µl 10 mM Tris (pH 7.5) or 50 µl nuclease free water. We then added 1 µl of 20 mg/ml RNase A and subsequently incubated at 65˚C for 30 min. DNA was then quantified using the Qubit Range dsDNA assay kit.

## Restriction digest of ancestral strain's barcode

Because over 90% of the initial population during the fitness measurement experiments consists of the ancestral strain, we sought to reduce the proportion of reads that represented its barcode to reduce sequencing costs. We thus performed restriction digestion using the ApaLI restriction site (GTGCAC) engineered into the barcode region of the DPY256 ancestral strain on DNA for each sample prior to (and following) PCR amplification. We added 1 µl of ApaLI (NEB #R0507L) and 5.5 µl of Cutsmart Buffer (NEB #R0507L) to genomic DNA and incubated at 37˚C for at least 1 h. Note that no barcode strains besides the ancestral strain contain this restriction site, due to the design of the barcode region.

## Barcode sequencing library preparation

To prepare sequencing libraries of the barcodes, we used a two-step PCR amplification protocol, as previously described [9,10,20]. In the first step of PCR, we use HPLC-purified primers that contain "inline indices" to label samples and 8-bp unique molecular identifiers (UMIs) to identify barcode reads from the same yeast cell that have been sequenced multiple times due to PCR amplification. Step 1 primer sequences are found in Table 1.

For the first step for PCR, we performed 8 or 16 reactions per sample, using approximately 4.8 µg (ranging between 3 µg and 7.5 µg) of DNA per sample across all reactions. Each set of eight 50 µl reactions included 16 µ of 50 mM MgCl2, 8 µl of 10 µm forward primer, 8 µl of 10 µm reverse primer, template DNA, and 200 µl of OneTaq HotStart 2X Master mix (NEB #M0484L). Three cycles of PCR was then carried out with the following steps:

**Table 1. PCR step 1 forward and reverse primers.**

| | |
|---|---|
| F201 | TCGTCGGCAGCGTC AGATGTGTATAAGAGACAG (N1:25252525)(N1)(N1) (N1)(N1)(N1) (N1)(N1)CGATGTT TAATATGGACTAAAGGAGGCTTTT |
| F202 | TCGTCGGCAGCGTC AGATGTGTATAAGAGACAG (N1:25252525)(N1)(N1) (N1)(N1)(N1) (N1)(N1)ACAGTGT TAATATGGACTAAAGGAGGCTTTT |
| F203 | TCGTCGGCAGCGTC AGATGTGTATAAGAGACAG (N1:25252525)(N1)(N1) (N1)(N1)(N1) (N1)(N1)TGACCAT TAATATGGACTAAAGGAGGCTTTT |
| F204 | TCGTCGGCAGCGTC AGATGTGTATAAGAGACAG (N1:25252525)(N1)(N1) (N1)(N1)(N1) (N1)(N1)GCCAATT TAATATGGACTAAAGGAGGCTTTT |
| F205 | TCGTCGGCAGCGTC AGATGTGTATAAGAGACAG (N1:25252525)(N1)(N1) (N1)(N1)(N1) (N1)(N1)ATCACGT TAATATGGACTAAAGGAGGCTTTT |
| F206 | TCGTCGGCAGCGTC AGATGTGTATAAGAGACAG (N1:25252525)(N1)(N1) (N1)(N1)(N1) (N1)(N1)CAGATCT TAATATGGACTAAAGGAGGCTTTT |
| F207 | TCGTCGGCAGCGTC AGATGTGTATAAGAGACAG (N1:25252525)(N1)(N1) (N1)(N1)(N1) (N1)(N1)GGCTACT TAATATGGACTAAAGGAGGCTTTT |
| F208 | TCGTCGGCAGCGTC AGATGTGTATAAGAGACAG (N1:25252525)(N1)(N1) (N1)(N1)(N1) (N1)(N1)TAGCTTT TAATATGGACTAAAGGAGGCTTTT |
| F209 | TCGTCGGCAGCGTC AGATGTGTATAAGAGACAG (N1:25252525)(N1)(N1) (N1)(N1)(N1) (N1)(N1)TTAGGCT TAATATGGACTAAAGGAGGCTTTT |
| F210 | TCGTCGGCAGCGTC AGATGTGTATAAGAGACAG (N1:25252525)(N1)(N1) (N1)(N1)(N1) (N1)(N1)ACTTGAT TAATATGGACTAAAGGAGGCTTTT |
| F211 | TCGTCGGCAGCGTC AGATGTGTATAAGAGACAG (N1:25252525)(N1)(N1) (N1)(N1)(N1) (N1)(N1)GATCAGT TAATATGGACTAAAGGAGGCTTTT |
| F212 | TCGTCGGCAGCGTC AGATGTGTATAAGAGACAG (N1:25252525)(N1)(N1) (N1)(N1)(N1) (N1)(N1)CTTGTAT TAATATGGACTAAAGGAGGCTTTT |
| R301 | GTCTCGTGGGCTCGG AGATGTGTATAAGAGACAG (N1:25252525)(N1)(N1) (N1)(N1)(N1) (N1)(N1)TATATACGC TCGAATTCAAGCTTAGATCTGATA |
| R302 | GTCTCGTGGGCTCGG AGATGTGTATAAGAGACAG (N1:25252525)(N1)(N1) (N1)(N1)(N1) (N1)(N1)CGCTCTATC TCGAATTCAAGCTTAGATCTGATA |
| R303 | GTCTCGTGGGCTCGG AGATGTGTATAAGAGACAG (N1:25252525)(N1)(N1) (N1)(N1)(N1) (N1)(N1)GAGACGTCT TCGAATTCAAGCTTAGATCTGATA |
| R304 | GTCTCGTGGGCTCGG AGATGTGTATAAGAGACAG (N1:25252525)(N1)(N1) (N1)(N1)(N1) (N1)(N1)ATACTGCGT TCGAATTCAAGCTTAGATCTGATA |
| R305 | GTCTCGTGGGCTCGG AGATGTGTATAAGAGACAG (N1:25252525)(N1)(N1) (N1)(N1)(N1) (N1)(N1)ACTAGCAGA TCGAATTCAAGCTTAGATCTGATA |
| R306 | GTCTCGTGGGCTCGG AGATGTGTATAAGAGACAG (N1:25252525)(N1)(N1) (N1)(N1)(N1) (N1)(N1)TGAGCTAGC TCGAATTCAAGCTTAGATCTGATA |
| R307 | GTCTCGTGGGCTCGG AGATGTGTATAAGAGACAG (N1:25252525)(N1)(N1) (N1)(N1)(N1) (N1)(N1)CTGCTACTC TCGAATTCAAGCTTAGATCTGATA |
| R308 | GTCTCGTGGGCTCGG AGATGTGTATAAGAGACAG (N1:25252525)(N1)(N1) (N1)(N1)(N1) (N1)(N1)GCGTACGCA TCGAATTCAAGCTTAGATCTGATA |

1. 94˚C for 10 min

2. 94˚C for 3 min

3. 55˚C for 1 min

4. 68˚C for 1 min

5. Repeat steps 2 to 4 twice for a total of 3 cycles

6. 68˚C for 1 min

7. Hold at 4˚C

The first-step PCR product was then column purified using the GeneJET Gel Extraction Kit (#K0692). Briefly, 100 μl of orange binding buffer were added to each 50 μl reaction. All 8 or 16 reactions from a given sample were pooled into the same purification column in a vacuum manifold. We then washed the column with 750 μl of wash buffer over vacuum. Then, each column was spun for 30 s at max speed to remove residual wash buffer. We then eluted into 47 μl of nuclease free water by centrifuging and stored the samples at 4˚C for the second step of PCR.

The second step of PCR further amplifies the barcodes and attaches Illumina indices as well as P5, P7 sequences for compatibility with Illumina sequencing, as done previously [20,21]. We used Nextera Index Xt v2 primers (Illumina #FC-131–2004) as found in Table 2.

**Table 2. PCR step 2 forward and reverse primers.**

| | |
|---|---|
| S513 | AATGATACGGCGACCACCGAGATCTACACTCGACTAGTCGTCGGCAGCGTC |
| S515 | AATGATACGGCGACCACCGAGATCTACACTTCTAGCTTCGTCGGCAGCGTC |
| S516 | AATGATACGGCGACCACCGAGATCTACACCCTAGAGTTCGTCGGCAGCGTC |
| S517 | AATGATACGGCGACCACCGAGATCTACACGCGTAAGATCGTCGGCAGCGTC |
| S518 | AATGATACGGCGACCACCGAGATCTACACCTATTAAGTCGTCGGCAGCGTC |
| S520 | AATGATACGGCGACCACCGAGATCTACACAAGGCTATTCGTCGGCAGCGTC |
| S521 | AATGATACGGCGACCACCGAGATCTACACGAGCCTTATCGTCGGCAGCGTC |
| S522 | AATGATACGGCGACCACCGAGATCTACACTTATGCGATCGTCGGCAGCGTC |
| N716 | CAAGCAGAAGACGGCATACGAGATTAGCGAGTGTCTCGTGGGCTCGG |
| N718 | CAAGCAGAAGACGGCATACGAGATGTAGCTCCGTCTCGTGGGCTCGG |
| N719 | CAAGCAGAAGACGGCATACGAGATTACTACGCGTCTCGTGGGCTCGG |
| N720 | CAAGCAGAAGACGGCATACGAGATAGGCTCCGGTCTCGTGGGCTCGG |
| N721 | CAAGCAGAAGACGGCATACGAGATGCAGCGTAGTCTCGTGGGCTCGG |
| N722 | CAAGCAGAAGACGGCATACGAGATCTGCGCATGTCTCGTGGGCTCGG |
| N723 | CAAGCAGAAGACGGCATACGAGATGGAGCGCTAGTCTCGTGGGCTCGG |
| N724 | CAAGCAGAAGACGGCATACGAGATCGCTCAGTGTCTCGTGGGCTCGG |
| N726 | CAAGCAGAAGACGGCATACGAGATGTCTTAGGGTCTCGTGGGCTCGG |
| N727 | CAAGCAGAAGACGGCATACGAGATACTGATCGGTCTCGTGGGCTCGG |
| N728 | CAAGCAGAAGACGGCATACGAGATTAGCTGCAGTCTCGTGGGCTCGG |
| N729 | CAAGCAGAAGACGGCATACGAGATGACGTCGAGTCTCGTGGGCTCGG |

Note that because of increased risk of index swapping associated with sequencing amplicons on Illumina machines with ExAmp technology [21], we labeled each sample with a unique combination of inline and Illumina indices. This allows for reads associated with index swapping due to mis-incorporation of indices or template swapping on the sequencing machine to be identified and removed from downstream analysis.

For the second step of PCR, we performed 3 reactions per sample. For each set of the 50 μl reactions, we used 45 μl of column purified Step 1 PCR product, 2.5 μl of the designated forward Nextera XT Index V2 primer (e.g., N716), 2.5 μl of the designated reverse Nextera XT Index V2 primer (e.g., S513), 3 μl of 10 mM dNTP (Fisher Scientific #PR-U1515), 1.5 μl of Q5 polymerase (NEB #M0491L), 30 μl of Q5 buffer (NEB #M0491L), and 65.5 μl of nuclease free water. We then ran the following program on the thermocycler to amplify for 20 cycles:

1. 98˚C 30 s

2. 98˚C 10 s

3. 62˚C 20 s

4. 72˚C 30 s

5. Repeat steps 2 to 4 19 times (20 cycles total)

6. 72˚C 3 min

7. Hold at 4˚C

We then performed column purification following a similar procedure to the purification from step 1, eluting instead into 30 μl of nuclease free water.

Following the second step of PCR, in order to further remove any residual ancestral barcode that were not digested before PCR amplification, we performed a second round of ApaLI

digestion, adding 3.5 μl of Cutsmart buffer and 1 μl of ApaLI restriction enzyme (NEB #R0507L) to each sample's Step 2 PCR product, digesting for at least 1 h at 37˚C. We then performed gel extraction using the GeneJet gel purification kit for each sample, keeping the 350 bp band representing intact barcode sequences. We then quantified the DNA concentration for each sample using Qubit HS kit (Thermo Fisher #Q32854), pooled such that each sample was equally represented in the final library and submitted for sequencing on Illumina sequencing machines.

## Tracking evolution

To track the dynamics of the evolution experiment, estimate the fitness of lineages during the evolution experiment, and infer the distribution of fitness effects, we extracted DNA and used PCR amplification to generate libraries for sequencing as described above, with the exception of not performing the ApaLI restriction digestion.

In order to identify barcode counts over time, we followed previously used custom scripts along with bartender (https://github.com/Sherlock-Lab/Barcode_seq/blob/master/bartender_BC1_BC2.py) to extract and cluster barcodes from timepoints along the evolution trajectory.

To infer fitness effects, the mean fitness of the population, and infer the from the evolution experiments themselves, we used FitMut1 [7,68]. To infer the distribution of fitness effects from this data, we used an approach developed in Levy and colleagues [7]. The general idea of this approach is to infer the distribution of fitness effects by counting the number of mutants arising with selection coefficients in the interval [s, s+ds] across the course of the evolution experiment. To infer a rate, we adjust the amount of time that this mutant could have arisen and been detected based on the mean fitness of the population, the time it takes for the mutant to establish, and its ability to rise to a detectable frequency in the population. Specifically, the number of mutations in the interval [s, s+ds] is expected to be:

$$\text{number of mutations in } ds = \mu(s)ds \times \left(\frac{s}{c}\right) \times N_e \int_0^{t-\left(\frac{1}{s}\right)\ln\left(\frac{n_0 s}{c}\right)} e^{-\bar{x}(t)}dt.$$

Where $N_e = 7 \times 10^7$ is the effective population size, $x(t)$ is the mean fitness of the population over time, $c \sim 3.5$ is the offspring number variance, and $n_0 \sim 1,000$ is the effective lineage size. We invert this function to estimate $\mu(s)$.

## Counting barcodes and calculating fitness from fitness measurement sequencing data

We used BarcodeCounter2 [9,69] to assign reads to their associated samples and barcodes. Briefly, we extracted the inline index, barcode, and UMI regions from each read using BLAST [70] to the known constraint region in the amplicon sequence. Then, we associated each read to its corresponding condition and time point based on its combination of Illumina and inline indices. We then used Bowtie2 [71] to map the extracted barcode regions to our known list of barcodes in the experiment, used UMIs to avoid over-counting duplicate reads, and counted the number of barcodes per sample.

To infer fitness values, we used the fitness inference procedure as developed previously. In each time interval, a mutant's fitness is calculated as its log-frequency change, adjusted by the mean fitness of the population. We infer the mean fitness of the population by calculating the log-frequency change of the set of 60 neutral lineages from Venkataram 2016.

## Frequency dependence

During the analysis of the fitness measurement data, we noticed a systematic shift of fitness over the course of the experiment, with many barcoded mutants showing a decline in fitness in 1- and 2-Day experiments as the fraction of the population that was adaptive increased, even after adjustment for changes in mean fitness (S1 Fig). These trends were not identified in previous experiments, and we suspect that this is due to frequency-dependent fitness effects driven by the very strongly adaptive mutants. To avoid the influence of these effects, we used only the first time point interval from 2-Day experiments (from time point 0 to time point 1), as this kept our fitness measurements consistent with previous studies [5,10,20,21]. Throughout the rest of the study, 2-Day fitness refers to this measurement using only early time points.

## Quantifying performances

To quantify mutant performances in each phase of growth, we quantified differences between fitnesses inferred from 1-, 2-, 3-, and 5-Day transfer experiments. Because the time interval between 24 and 48 h only contains respiration phase, we quantified respiration performance per hour as:

$$\text{ResPerHour} = \text{2-Day fitness} - \text{1-Day fitness}/24 \text{ hrs}$$

To calculate fermentation performance, we removed the 4 hours worth of respiration performance from the 1-Day fitness and divided the remaining fitness into the 16 hours of fermentation performance (accounting for approximately 4 h of lag phase):

$$\text{FerPerHour} = (\text{1-Day fitness} - 4*\text{ResPerHour})/16 \text{ hrs}$$

Because 1-Day fitness measurements are used for both respiration and fermentation performances, there is the potential for noise in 1-Day measurements to introduce a relationship between fermentation and respiration performances. To eliminate measurement noise from having this influence, we used different replicates of the 1-Day fitness to calculate fermentation and respiration performance. Specifically, we used the replicate 2 flasks to calculate respiration performance and the replicate 1 flasks to calculate fermentation performance.

To infer stationary phase performance, we took the difference between 5- and 3-day fitness and divided by 48 h of time:

$$\text{StaPerHour} = (\text{5-Day fitness} - \text{3-Day fitness})/48 \text{ hrs}$$

To calculate the uncertainty of performances, we used error propagation from the estimated errors of fitness. To calculate performances relative to parental strain, we computed the difference between each mutant's performance and its parental strain. For *CYR1*, *GPB2*, *TOR1*, and *IRA1-missense* second-step mutations, we used the mean of the neutral barcode strains as the parental reference measurement. For *IRA1-nonsense* second-step mutations, for which no neutral clones were isolated, we used the parental barcoded mutant present in the pool of first-step mutants (denoted with a "+" in main text figures).

To calculate the uncertainty in the proportion of mutants that improve fermentation and respiration or other performances as depicted in Fig 3C, we use a re-sampling test based on the uncertainty of performances derived from measurement error above. For example, to calculate the uncertainty in the percentage of mutants from the second step that improve both fermentation and respiration performance, we conducted 1,000 re-samplings of all mutant performances according to measurement uncertainty. We then computed the percentage of mutants that improved both fermentation and respiration in this resampling. We repeated this

computation for each re-sampling and reported the standard deviation in the percentage of mutants that are in each quadrant.

## Differences in selection pressure do not drive the shift towards modular adaptation

To evaluate whether a systematic shift in selection pressure occurred during the second-step evolution experiments, we identified mutants for which we called their evolution fitness from the estimation of the distribution of fitness effects. Because many of the remaining mutants are pure diploids whose spread may be dominated by measurement noise, we removed these mutants from the list. This resulted in a set of 185 second-step mutants. We then initially performed an analysis on mutants isolated from the IRA1-missense parental strain, which had the most (82) second-step adaptive mutants identified in this manner. We found a high correlation between the fitness measured in our barcoded fitness measurement experiments and the estimated fitness from the evolution experiment (Pearson correlation of 0.7). We also plotted respiration and fermentation performances to understand whether there are systematic differences in the contribution of respiration and fermentation to evolution fitness compared to the fitness measurement experiments (S6 Fig). To explicitly test for such a systematic shift in selection pressure, we performed a partial correlation analysis. If respiration contributes more to fitness in the evolution condition, we would expect to see a correlation between respiration performance and evolution fitness, after correcting for fitness in the measurement experiment. We find no such correlation (r = −0.01, $p$ = 0.926). We similarly see no correlation between fermentation performance and evolution fitness after performing the same correction (r = 0.04, $p$ = 0.715).

We performed a similar partial correlation analysis for the full set of 185 second-step adaptive haploids for which evolution fitnesses were estimated, finding no relationship between respiration performance and evolution fitness, accounting for fitness measurement fitness (r = −0.02, $p$ = 0.74) nor a relationship between fermentation performance and evolution fitness after accounting for fitness measurement fitness (r = 0.04, $p$ = 0.613).

## 1-Day evolution experiment analysis

To evaluate whether yeast adapting to a 1-Day transfer could further improve their fermentation performance, we quantified the performance of 1-Day mutants as above. We identified several mutants with fermentation performances meeting or exceeding the maximum fermentation performance achieved by first-step mutants. Using a threshold of at least 2 standard errors (which corresponds to an FDR of $p$ < 0.05), a single second-step mutant that arose in the Evo1D *IRA1*-nonsense population had fermentation performance that exceeded the first-step maximum.

## Whole genome sequencing

We selected mutants for whole genome sequencing based on their fitness and performance in the growth phase, such that we selected as many unique mutants as possible based on their performances and those that had barcodes confidently identified by the metagrid. This resulted in a total of 346 clones targeted for sequencing.

Clones that were selected for sequencing were grown in 500 μl of YPD in 96-well deep well plates for 2 days at 30°C without shaking, and 400 μl of saturated culture was collected from each well for genomic DNA extraction using the Invitrogen PureLink Pro 96 Genomic DNA Kit. Libraries were prepared using a ⅕ dilution protocol of the Illumina DNA prep, using Illumina Unique Dual Indexing primers.

## Variant calling

To identify variants from the sequencing data, we used bwa [72] to align all reads to the S288C reference genome (R64-1-1-20110203). We then used picard (https://broadinstitute.github.io/picard/) to fix read groups and marked duplicate reads. We then used GATK (version 4.2.0.0) [73] to generate individual GVCF files, merge GVCF files, and call genotypes on all samples. After removing samples with less than 20× coverage, we removed variants according to the following filters: QD < 5, FS < 60, SOR < 3, MQ < 50, MQRankSum < -3.0, ReadPosRankSum < -5.0. After this filtering, we further removed ancestral variants present in all samples, mitochondrial variants, variants with GQ less than 70. This filtering resulted in 727 sites that were variable across our samples. We then manually inspected all called variants, resulting in 631 manually verified variants. We then used bcftools [74] to filter the vcf file to these verified variants and used snpEff [75] to annotate variants.

We then assigned variants to the corresponding barcoded mutants based on plate position. To check that our assignment was correct, we also verified the barcodes from the whole genome sequencing reads. For the 326 mutants for which we had sufficient coverage of the barcode region (at least 4 successfully mapped barcode reads), 324 had the correct barcode identified. We opted to not use sequencing information from the 2 samples with mismatching barcodes between the sequencing and expected based on clone isolation barcode sequencing.

We further identified preexisting mutations in which identical mutations were present in several sequenced mutants of a given low-complexity barcode. These mutations were classified as "preexisting" mutations and ignored in downstream analyses except in cases where they belonged to a putatively causal gene (see "Mutation and ploidy classification" section). See supplementary data sheet S2 Data for the list of preexisting mutations and verbose characterization of the called mutations.

## Mutation and ploidy classification

To identify mutations likely responsible for driving fitness gains in these experiments, we identified putative adaptation-driving mutations by identifying mutations that occurred in genes that were recurrently mutated across adaptive clones. Specifically, genes with 4 or more mutations were classified as likely adaptation-driving. After classifying genes based on their function, we further identified additional mutations as adaptation-driving due to their effect on similar processes as recurrently mutated genes.

To classify the ploidy of mutations, we initially classified mutants according to their performance in the benomyl assay. We additionally classified mutants as "pure diploids" and "neutral haploids" by their similarity to the large cluster of haploids and diploids in terms of their fitness effects across all the conditions. Mutants that were within this large cluster of diploids but initially classified as haploids according to the benomyl assay were classified as pure-diploids.

From this initial ploidy classification, the majority of mutants which exhibited mutations in *PAB1* were classified as diploids, perhaps reflecting a sensitivity of *PAB1* mutants to benomyl. We re-classified all *PAB1* mutants as adaptive haploids with respiration performance relative to parental strain less than 0.06. *PAB1* mutants with greater respiration performance were classified as high-fitness diploids, consistent with the effect that auto-diploidization had on mutations from other genes. Similarly, *PAN2* and *PAN3* mutants were classified as diploids and have previously been shown to be susceptible to benomyl [76]. Given we had few of these mutations, we did not have enough information to reclassify these mutations as we did for *PAB1*.

We also noticed that the "neutral haploids" for *GPB2* and *TOR1* parental strains differed in respiration performance from the parental strain present in the collection of first-step mutants they were originally derived from. This difference may be due to additional mutations that arose during the barcoding process. In particular, *GPB2* mutants harbor an additional LPD1 downstream variant and *TOR1* mutants harbor an additional *MRPL10* missense variant (see Variant calling).

## Supporting information

**S1 Fig. Evidence for frequency dependence in fitness measurement experiments.** The vertical axis of each subplot depicts the percent deviation from Li 2019 fitness values for the set of adaptive haploids that were present in Li 2019 fitness measurements and this study. The horizontal axis is the fraction of the population that is adaptive. Points show the deviation for each mutant, with the median across all mutants depicted by the heavy circle. Blue and orange points are from experiments initiated with the adaptive barcode pool consisting of 2% and 5% of the population, respectively. Red dotted line indicates the deviation for the overall fitness measurement used throughout the paper. Red box in (B) refers to the time points used. Subpanels A–D refer to Fit1D, Fit2D, Fit3D, and Fit5D fitness values, respectively. The data and code underlying this figure can be found in https://zenodo.org/records/13336585.
(TIF)

**S2 Fig. 1-Day evolution experiments identify mutants that improve fermentation performance.** Fermentation and respiration performances for mutants discussed in the main text and Evo1D mutants (in red). Despite less dense sampling, we find at least 1 Evo1D mutant (indicated with red arrow) with fermentation performance that exceeds the highest fermentation performance from first-step mutants (blue vertical line). Error bars on mutant with high fermentation performance denote 2 standard deviations of measurement error. The data and code underlying this figure can be found in https://zenodo.org/records/13336585.
(TIF)

**S3 Fig. Fermentation and respiration phase performances by parental strain with uncertainty.** Each subpanel depicts a scatter plot with the fermentation and fermentation performances for each parental strain as in Fig 3. Lighter points indicate Evo2D mutants, darker points indicate Evo3D mutants. Error bars denote 2 standard deviations of measurement error per point. The data and code underlying this figure can be found in https://zenodo.org/records/13336585.
(TIF)

**S4 Fig. Fermentation and stationary phase performances by parental strain.** Each subpanel depicts a scatter plot with the fermentation and stationary performances for each parental strain. Lighter points indicate Evo2D mutants, darker points indicate Evo3D mutants. The data and code underlying this figure can be found in https://zenodo.org/records/13336585.
(TIF)

**S5 Fig. Respiration and stationary phase performances by parental strain.** Each subpanel depicts a scatter plot with the respiration and stationary performances for each parental strain. Lighter points indicate Evo2D mutants, darker points indicate Evo3D mutants. The data and code underlying this figure can be found in https://zenodo.org/records/13336585.
(TIF)

**S6 Fig. Comparison of fitness measurement and evolution fitness.** Each subpanel depicts a scatter plot comparing the measured Fit2D fitness on the x-axis and the estimated fitness from

the evolution trajectories for Evo2D IRA1-missense mutants. **(A)** Mutants colored by gene. **(B)** Mutants colored by respiration performance. **(C)** Mutants colored by fermentation performance. The data and code underlying this figure can be found in https://zenodo.org/records/13336585.
(TIF)

**S7 Fig. Molecular targets of adaptation by gene.** Performance effects of mutations separated by biological process or pathway as in Fig 5. Points are colored by gene, and shape indicates ploidy (circles are haploids, triangles diploids). Kernel Density Estimates show density of neutral haploids for each parental strain (solid lines) and pure diploids for each parental strain (dashed lines). **(A)** First-step mutants. **(B)** Second-step mutants depicted, with performances measured relative to parental strain. The data and code underlying this figure can be found in https://zenodo.org/records/13336585.
(TIF)

**S8 Fig. Molecular targets of adaptation by gene.** Colored by gene, shape depicts ploidy (circles are haploids, triangles diploids). Kernel Density Estimates show density of neutral haploids for each parental strain (solid lines) and pure diploids for each parental strain (dashed lines). The data and code underlying this figure can be found in https://zenodo.org/records/13336585.
(TIF)

**S9 Fig. Mutational effects on fermentation and stationary phase performance.** Colored by gene, shape depicts ploidy (circles are haploids, triangles diploids). Kernel Density Estimates show density of neutral haploids for each parental strain (solid lines) and pure diploids for each parental strain (dashed lines). The data and code underlying this figure can be found in https://zenodo.org/records/13336585.
(TIF)

**S10 Fig. Mutational effects on respiration and stationary phase performance.** Colored by gene, shape depicts ploidy (circles are haploids, triangles diploids). Kernel Density Estimates show density of neutral haploids for each parental strain (solid lines) and pure diploids for each parental strain (dashed lines). The data and code underlying this figure can be found in https://zenodo.org/records/13336585.
(TIF)

**S1 Data. Performance and mutation data for all isolated barcoded mutants.**
(XLSX)

**S2 Data. Mutation calling information for whole genome sequencing.**
(XLSX)

**S3 Data. Performance, fitness, and inferred evolution fitness for all isolated barcoded mutants.**
(CSV)

## Acknowledgments

The authors would like to thank Kerry Geiler-Samerotte, Olivia Ghosh, and members of the Petrov and Sherlock labs for helpful comments and discussion. We would like to thank Stanford University and the Stanford Research Computing Center for providing computational resources and support that contributed to these research results.

## Author Contributions

**Conceptualization:** Grant Kinsler, Yuping Li, Gavin Sherlock, Dmitri A. Petrov.

**Formal analysis:** Grant Kinsler, Yuping Li.

**Funding acquisition:** Gavin Sherlock, Dmitri A. Petrov.

**Investigation:** Grant Kinsler, Yuping Li.

**Methodology:** Grant Kinsler, Yuping Li.

**Software:** Grant Kinsler, Yuping Li.

**Supervision:** Gavin Sherlock, Dmitri A. Petrov.

**Visualization:** Grant Kinsler.

**Writing – original draft:** Grant Kinsler.

**Writing – review & editing:** Grant Kinsler, Yuping Li, Gavin Sherlock, Dmitri A. Petrov.

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
