## [Editor Report · Decision Letter 0]

26 Apr 2024

Dear Dr Kinsler, 

Thank you for submitting your manuscript entitled "A shift from pleiotropic to modular adaptation revealed by a high-resolution two-step adaptive walk" for consideration as a Research Article by PLOS Biology.

Your manuscript has now been evaluated by the PLOS Biology editorial staff, as well as by an academic editor with relevant expertise, and I'm writing to let you know that we would like to send your submission out for external peer review.

Once your full submission is complete, your paper will undergo a series of checks in preparation for peer review. After your manuscript has passed the checks it will be sent out for review. To provide the metadata for your submission, please Login to Editorial Manager (https://www.editorialmanager.com/pbiology) within two working days, i.e. by Apr 30 2024 11:59PM.

Kind regards,

Roli Roberts

Roland Roberts, PhD

Senior Editor

PLOS Biology

rroberts@plos.org

---

## [Decision Letter · Decision Letter 1]

27 Jun 2024

Dear Grant,

Thank you for your patience while your manuscript "A shift from pleiotropic to modular adaptation revealed by a high-resolution two-step adaptive walk" went through peer-review at PLOS Biology. Your manuscript has now been evaluated by the PLOS Biology editors, an Academic Editor with relevant expertise, and by three independent reviewers.

You’ll see that reviewer #1 says that he enjoyed the manuscript, but raises two issues with the interpretation and implications; one is about the framing around the “cost of complexity” hypothesis, and the other is the generalisability of the findings beyond the system under study. He has a number of other textual and/or presentational points. Reviewer #2 likes the study and the paper, but asks about a potential confound and requests several clarifications. Reviewer #3 is also positive, and really only has one significant question, which is related to reviewer #1’s point about generalisability, and draws on lessons learned from the LTEE. We discussed these comments with the Academic Editor, who also asked me to pass on some additional guidance to you (see foot of this email).

In light of the reviews, which you will find at the end of this email, we are pleased to offer you the opportunity to address the comments from the reviewers in a revision that we anticipate should not take you very long. We will then assess your revised manuscript and your response to the reviewers' comments with our Academic Editor aiming to avoid further rounds of peer-review, although might need to consult with the reviewers, depending on the nature of the revisions.

IMPORTANT:

a) Please change your title to something more explicit and engaging for our broader readership. We suggest something like: "A high-resolution two-step evolution experiment in yeast shows that mutations shift from pleiotropic to modular adaptation"

b) Please address the concerns raised by the reviewers, AND the additional comments from the Academic Editor (see foot of this email).

c) Please ensure that you comply with PLOS' Data Policy; specifically, we need you to supply the numerical values underlying Figs 2DEF, 3ABC, 4AB, 5ABC, S1ABCD, S2, S3, S4, S6, S6ABCDEF, S7ABCDEF, S8ABCDEF, either as a supplementary data file or as a permanent DOI’d deposition. I note that you already have an associated GitHub deposition and a raw-looking supplementary data file. Please ensure that the data and code required to generate the Figs are made available. Also, because Github depositions can be readily changed or deleted, please make a permanent DOI’d copy (e.g. in Zenodo) and provide this URL (see below).

d) Please cite the location of the data clearly in all relevant main and supplementary Figure legends, e.g. “The data underlying this Figure can be found in S1 Data” or “The data underlying this Figure can be found in https://zenodo.org/records/XXXXXXXX

e) Please make any custom code available, either as a supplementary file or as part of your data deposition.

**IMPORTANT - SUBMITTING YOUR REVISION**

*Resubmission Checklist*

*Published Peer Review*

*PLOS Data Policy*

*Blot and Gel Data Policy*

Sincerely,

Roli

Roland Roberts, PhD

Senior Editor

PLOS Biology

rroberts@plos.org

REVIEWERS' COMMENTS:

Reviewer #1:

[identifies himself as Alex Couce]

In this manuscript, Kinsler and cols. couple experimental evolution with high-resolution barcode tracking to study pleiotropy patterns among adaptive mutations in a yeast under glucose-limited, batch culture conditions. To break down fitness gains into individual trait improvements (thus quantifying pleiotropy), they leverage the specific physiology of yeast in these conditions, in which they experience two sequential growth phases: first, fermenting glucose to ethanol, and then respiring the ethanol produced in the previous phase. This growth cycle, combined with high-resolution barcode tracking, allows for a detailed analysis of pleiotropy, both in terms of abundance and sign (i.e., whether improvements in one phase are correlated, anticorrelated, or show no correlation with the other phase). In a prior work, the authors tracked the frequency changes of hundreds of mutants generated in the ancestral strain across both the fermentative and respiratory phases, revealing that the vast majority of the mutants improved performance in both phases, not just in one. Here, they conduct a follow-up evolution experiment to examine the distribution of pleiotropic effects among adaptive mutations generated in five different backgrounds, each carrying a single beneficial mutation acquired in the prior work. In contrast to their observations with the ancestral strain, they report that in this second adaptive step, most of the new beneficial mutations improve performance in only a single growth phase (respiration). Consistent with this drop in the abundance of adaptively pleiotropic mutations, they show that the molecular targets of selection also shift from glucose-sensing pathways to targets related to mitochondrial function and respiration. They argue that this shift from pleiotropic to modular phenotypic effects may be a general phenomenon during the initial steps of adaptation to novel selective pressures. In particular, they hypothesize that sensing pathways might have been shaped during past evolutionary history to coherently affect multiple traits at once, especially traits important for handling concomitant selective pressures.

Overall, I enjoyed the manuscript. The questions posed are relevant, and their methodology provides a uniquely sound way of exploring them. I only have two issues that have to do with the interpretation of their findings and their implications within the broader context of the field. First, the way the manuscript is framed from the start (opening of the abstract and the introduction), it reads as if it is going to be a rebuttal of Orr's 2000 "cost of complexity" hypothesis, the reduction in adaptation rate imposed by pleiotropy. This is problematic because, for one thing, later work predicts that such cost should either be much less substantial or even absent (Wang, PNAS 2010, PMID: 20876104; Wagner, Nature 2008, PMID: 18368117). In addition, and even leaving a proper account of the theory aside, I think it is important to emphasize that the present work's results indicate that a core assumption of Orr's model — that pleiotropy is forcibly random — is inaccurate, rather than suggesting that Orr's predictions are incorrect. In fact, the shift to non-pleiotropic and small-effect adaptive mutations can be seen as consistent with these predictions. I'd encourage the authors to elaborate more in detail how the 'cost of complexity' hypothesis and its implications are affected by their findings.

Second, while I like the idea that signaling pathways may serve as high-leverage routes for improving many traits simultaneously, I do not think at this point we have enough evidence to determine whether this is a general phenomenon or, in contrast, whether it is just a peculiarity of Ras/PKA mutations in yeast undergoing two consecutive growth phases in glucose-limited batch cultures. In fact, the other example of a first-step mutation involving a signaling pathway shown by the authors is clearly modular (TOR/Sch9a), improving fermentation but not respiration. To be clear, I find the idea plausible, and in fact I believe it has been partly anticipated in earlier suggestions that adaptive pleiotropy could emerge due to selection on co-varying traits (Cheverud 1996, doi.org/10.1093/icb/36.1.44). However, I missed a discussion on the prevalence of adaptive pleiotropy examples in other model systems. For instance, in Lenski's Long Term Evolution, it is known that the two primary traits under selection are the shortening the lag phase and the increase of the exponential growth rate (Vasi 1994, doi.org/10.1086/285685). Several large effect mutations appear in parallel across the replicate populations in the first step of adaptation, including mutations in regulatory genes (fis, nadR, spoT, topA). Of these, only spoT seems to be a good candidate for pleiotropically improving the two traits under selection, while others are presumably much more modular, improving only growth rates (e.g. topA) (Cooper, 2003; PMID: 12538876). I think the greater value of the present work lies not in suggesting that adaptive pleiotropy may be general, but in demonstrating that it is more common than previously anticipated. As a consequence, I believe that explicitly stating the knowledge gap in the prevalence of adaptive pleiotropy, rather than (or in addition to) just speculating about its presumed generality in signaling networks, will enhance the overall impact of the work by encouraging others to explore these questions in their model systems.

Minor points:

- Abstract: "Our results suggest that the genes in cellular signaling pathways are particularly capable of providing large, adaptively pleiotropic benefits to the organism due to their ability to coherently affect many phenotypes at once." <- As explained above, at this point, with only one example in favor (Ras/PKA) and one against (TOR/Sch9a), this statement comes across as too strong. Consider phrasing it to better reflect the hypothetical nature of the idea: "_may be more likely to provide_ large, adaptively pleiotropic benefits"

- Figure 2C: Since much of the discussion is about whether adaptive pleiotropy is a general property of signaling systems, it'd be useful to clarify here that four of the five first-step mutations involved the Ras/PKA signaling patwhay (CYR1, GPB2, both IRA1), while the remainder involves the TOR/Sch9 one. Alternatively, this may be indicated on the upper side of Table 1.

- Fig 3C: I missed an explanation in the legend about the meaning of the red arrows, as well as a reminder that, compared to panel B, we are now looking at values relative to the ancestor. Also, please spell out "KDE" in "KDE estimates".

L130-132: "This suggests that these populations have not yet reached physiological constraints but rather that adaptive walks may be constrained by genetic modules which prevent adaptive mutations from improving multiple performances in a single step". I got stuck in this convoluted sentence and had to read the whole paper to grasp its meaning (i.e., that genetic constraints matters more than physiological constraints?). Please consider rewriting, it could be perfectly broken down into 2-3 sentences introducing and explaining the different concepts involved.

L205: I think it'd be helpful to explicitly state here that the ancestral strain is not barcoded and represents >90%.

L237: "selection coefficients per cycle". Please consider giving at least an intuition of number of generations per cycle to allow comparisons with other experimental systems.

L262: "fitness benefits of 55%, 48%, and 52%". Are these differences significant?

L302: please clarify the meaning a purpose of "re-sampling test", I'd assume it refers to the precision of the estimates?

L330-31: "only a small number of mutants have fermentation performances worse than the original ancestor strain (Figure 3C, vertical black dashed line in each subplot)". Since there are no error bars or the like, please consider providing some indication of the uncertainty surrounding these values, as well as the extent to which the statement is robust.

L344-46: "This mutant does improve performance in both fermentation and respiration growth phases, albeit with most of its fitness gain coming from respiration". Same as above, please consider providing some indication of the uncertainty surrounding these values, as well as the extent to which the statement is robust. In both cases, I understand the plots are already complex to add error bars, but perhaps a figure could be added to the SI to show the magnitude of measurement errors. 

L351: "some of these very fit clones have acquired third or fourth adaptive steps". That's pretty neat. Have you checked whether these clones are mutators?

L396: "42% of second-step Evo2D adaptive mutants have lower stationary performance than their parental strain". Please clarify whether this value refers only to points with measurement errors that does not overlap 0 (as later mentioned in L443-444).

L490: "we isolated at least one second-step mutation from the IRA1-nonsense population that improved fermentation performance above the highest fermentation performances achieved by first- or second-step mutations evolved in the 2-Day and 3-Day environments (see Fig S2). This suggests that while a fermentation performance maximum has not yet been reached, the pre-existing wiring of genetic and signaling pathways may be such that...". Again, please consider providing some indication of the uncertainty surrounding these values, as well as the extent to which the statement is robust.

L493: "This suggests that while a fermentation performance...". Please consider reminding the reader that this is the third possibility you anticipated in line 463 above.

L701: "we sought to understand the frequent observation that single adaptive mutations observed in experimental evolution, especially those of large effect, can pleiotropically improve multiple distinct performances at once. This observation is puzzling because theoretical work suggests that as pleiotropy increases, large effect adaptive mutations should become less probable". -> I am not sure I agree with the way the sentence reads. For one thing, examples of pleiotropically adaptive mutations are scarce in the literature. It is true that a consistent pattern emerging from multiple studies is that the fist steps of adaptation typically involve genes encoding global regulators. And while these mutations are pleiotropic, in most cases there is no indication that this pleiotropy is adaptive in the sense of simultaneously improving several adaptive traits at once (Hindré, 2012; PMID: 22450379). I believe the major takeaway of the present work is precisely the demonstration that large effect, pleiotropically adaptive mutations exist, rather than "understanding" a phenomenon that was not even clearly demonstrated before. In addition, "puzzling" sounds rather strong to me. Please specify this refers to Orr's work, and note that later work predicts that the cost of pleiotropy should either be much less substantial or even absent (Wang, PNAS 2010, PMID: 20876104; Wagner, Nature 2008, PMID: 18368117).

Reviewer #2:

The manuscript by Kinsler and co-authors explores the role of pleiotropy in early adaptation. While experimental evolution studies have shown that early steps in adaptation are often pleiotropic, resulting in large fitness gains, this phenomenon has not been meticulously investigated. The authors performed evolution experiments starting from five first-step mutants identified previously. They measured the fitness of hundreds of second-step mutants and compared these to the fitness effects of the first-step mutants, indicating that initial large-effect pleiotropic adaptation shifts to more modular adaptation with on average smaller, but still significant, effect sizes. The authors then go on to show that this goes along with a shift in mutational targets involved in signaling pathways towards targets involved in respiration and mitochondrial function.

Overall, the study addresses an important question in evolutionary biology regarding the nature of adaptive mutations and the fascinating paradox of the theoretical "cost of complexity" contrasted with the fact that we often observe pleiotropic mutations in experimental evolution. Their findings provide valuable insights into the dynamics of evolution and hold significant general interest. The paper is well-written and their results are beautifully presented.

To determine the respiration, fermentation, and stationary performance of the mutants, the authors examined the difference between 2-Day and 1-Day fitness, the first 20 hours, and the difference between 5-Day and 3-Day fitness, respectively. However, I'm concerned about the possibility of shifts in growth rates, leading to temporal changes in the transition from fermentation to respiration or respiration to the stationary phase in mutants. These shifts could potentially impact how accurately the different fitness measurements correspond to their intended growth phases. Have the authors investigated this potential confounding issue?

To determine whether the shift towards modular adaptation was not induced by a simultaneous shift in the selection pressure towards enhancing respiration, the authors conducted a partial correlation between respiration performance and fitness during the evolution experiment, while adjusting for the fitness inferred from their fitness measurements. I must admit, I'm somewhat lost regarding this analysis, and a clearer explanation would greatly aid in understanding its significance. Could the authors offer further clarification on this analysis, as I find that indicating that their wasn't a shift in selection pressure pivotal to their conclusion?

On average, how many mutations per mutant were identified in the different Evo experiments, and was this similar to the first-step Evo experiment? Were there for example on average more mutations per clone in the first step evo experiment compared to the second step experiments?

The authors propose that the observed shift towards modular adaptation is driven by the exhaustion of mutation targets in signaling pathways. If I understand correctly, this is mainly due to the absence of second-step adaptive mutations in signaling pathways. However, the observation of second-step mutations targeting specific improvements in respiration performance does not necessarily indicate the absence of any signaling pathway targets. Rather, the effect size of the pleiotropic mutations becomes comparable smaller, because of the cost of complexity/pleiotropy. I.e. the absence of these mutations is simply due to the comparatively larger effect sizes of the respiration-specific mutations. This is supported by the Evo1D experiment, which demonstrates that improvement in fermentation is still achievable. Therefore, it seems that the targets in signaling pathways are still present, and that it's not the exhaustion but rather just the larger effect sizes of respiration targets.

Line 731" These results support a model of adaptation wherein early adaptation is driven by mutations of large effect that improve multiple performances at once. Then, after these mutations have become exhausted, adaptation may proceed via more modest mutations that improve performances in a stepwise manner": I don't think their results seem to support this generalized scenario. All five second-step conditions (although the IRA1 nonsense maybe to a lesser extent) still show the presence of large effect second-step mutations, comparable to top first-step effect sizes. It's important here to note that it's on average their scenario might be the case, but depending on the conditions, for example in situations with strong clonal interference, large effects mutations will preferentially be selected for.

The authors mention that they performed evolution experiments in replicate, but it is unclear how they take the resulting variation into account in their downstream analyses. Did both replicate population behave similarly?

Minor comments:

The colors differentiating Evo2D and Evo3D weren't always clearly distinguishable (Fig 2EF, 3C).

Why aren't the crosses in Fig 3C within the neutral isoclines? Wouldn't this be the expectation?

How many generations or transfers were conducted for Evo1D and Evo3D experiments? I couldn't find this info.

Fig 2EF: Slanted X-labels are too small.

Reviewer #3:

[identifies himself as Christopher Marx]

This paper tackles a key issue in terms of adaptive trajectories: is there a particular commonality to the manner in which beneficial mutations generate their fitness effects? Using a very well-characterized yeast system and a super powerful implementation of barcodes, they show rather convincingly that, in their two (or three) phase environments the first beneficial mutations showed synergistic pleiotropy with single mutations affecting both fermentation and respiration, the second mutations were decidedly more modular and changed one or the other and very rarely both. All of this is very well done and clearly written.

I only have one major concern: how general is this? This is less a concern about it being just one environment and one model system, as that is true for any experimental system. It is more that - as the authors state - the vast majority of first step mutations were in the Ras/PKA pathway. Does this open the door to synergistic pleiotropy in a way that is unusual? Does this underlie the "regulation then mitochondria" ordering of mutations?

Knowing what the field does from other systems, this is already known not to be far from universal. One of the Lenski lines first had a mutation deleting the ribose operon, topoisomerase, ppGpp synthase, a uridyltransferase, and then pyruvate kinase. The ppGpp synthase affects a global secondary messenger, but the others are not global regulators in the traditional sense (even if they do have broad effects). Numerous other adaptive trajectories are known from experimental evolution so the opportunity to do at least some degree of comparison is available, at least at the level of gene annotations.

I feel what is needed is simply a bit more reach to other systems and a step back from making too big of claims of generality. It may very well come to be true that the biggest effect, first step mutations do tend to have synergistic pleiotropy and by the second mutation shift to being modular, but I think much more data is needed before making too strong of a claim.

Christopher Marx

COMMENTS FROM THE ACADEMIC EDITOR (lightly edited):

Besides the specific questions of rev#1 and especially rev#2 about interpretation of the results, the reviewers seem to agree that the authors should be asked to (i) better acknowledge previous work on modular pleiotropy (by Wagner and others), and (ii) particularly better discuss the generality of their findings, i.e. the decreasing pleiotropy of mutations during adaptation and the role of signalling pathways. In addition to these comments by the reviewers, I think the authors should also address the possible role of mutation bias, i.e. to which extent are first-step mutations driven by high mutation rates in stead of large fitness benefits?

I suggest that the authors, for their discussion of the generality of their findings, have a look at a recently published meta-analysis. Ruelens et al. 2023 (doi:10.1098/rstb.2022.005) report a meta-analysis of the pleiotropy of mutations in lab evolution experiments with E. coli, including Lenski's LTEE and 23 other studies. The analysis shows that point mutations (but not indels or large deletions or duplications, because they cannot realize the combinations of pleiotropic effects underlying these large benefits!) in highly pleiotropic genes (measured by the number of protein-protein interactions) are among the largest-benefit mutations under constant lab conditions. These findings for E. coli may therefore provide a more general explanation for the authors’ observation, that first-step mutations in yeast are more pleiotropic than later mutations – because clonal interference will promote large-effect mutations to be fixed first.

---

## [Editor Report · Decision Letter 2]

17 Sep 2024

Dear Dr Kinsler,

Thank you for the submission of your revised Research Article "A high-resolution two-step evolution experiment in yeast reveals a shift from pleiotropic to modular adaptation" for publication in PLOS Biology. On behalf of my colleagues and the Academic Editor, Arjan de Visser, I'm pleased to say that we can in principle accept your manuscript for publication, provided you address any remaining formatting and reporting issues. These will be detailed in an email you should receive within 2-3 business days from our colleagues in the journal operations team; no action is required from you until then. Please note that we will not be able to formally accept your manuscript and schedule it for publication until you have completed any requested changes.

IMPORTANT: I've asked my colleagues to include the following request alongside their own: It seems unlikely that S1 Data is sufficient to allow the reproduction of the Figs. Your Zenodo deposition (https://zenodo.org/records/13336585) may be a more resource to direct readers to. Please clarify and/or change the citation in each of your Figure legends to "The data and code underlying this Figure can be found in https://zenodo.org/records/13336585."

Sincerely, 

Roli Roberts

Senior Editor

PLOS Biology

rroberts@plos.org